# Direct neurotransmitter activation of voltage-gated potassium channels

Rían W. Manville[1], Maria Papanikolaou[1] & Geoffrey W. Abbott[1]

Voltage-gated potassium channels KCNQ2–5 generate the M-current, which controls neuronal excitability. KCNQ2–5 subunits each harbor a high-affinity anticonvulsant drug-binding pocket containing an essential tryptophan (W265 in human KCNQ3) conserved for >500 million years, yet lacking a known physiological function. Here, phylogenetic analysis, electrostatic potential mapping, in silico docking, electrophysiology, and radioligand binding assays reveal that the anticonvulsant binding pocket evolved to accommodate endogenous neurotransmitters including γ-aminobutyric acid (GABA), which directly activates KCNQ5 and KCNQ3 via W265. GABA, and endogenous metabolites β-hydroxybutyric acid (BHB) and γ-amino-β-hydroxybutyric acid (GABOB), competitively and differentially shift the voltage dependence of KCNQ3 activation. Our results uncover a novel paradigm: direct neurotransmitter activation of voltage-gated ion channels, enabling chemosensing of the neurotransmitter/metabolite landscape to regulate channel activity and cellular excitability.

[1] Department of Physiology and Biophysics, Bioelectricity Laboratory, Medical Sciences D, ZOT 4560, School of Medicine, University of California, Irvine, CA 92697, USA. Correspondence and requests for materials should be addressed to G.W.A. (email: abbottg@uci.edu)

K⁺ efflux is the primary force behind the cellular repolarization that ends action potentials, and is also important in setting and maintaining resting membrane potential, and preventing aberrant action potential firing. In vertebrate nervous systems, the M-current, a muscarinic-inhibited Kv current[1] generated by KCNQ (Kv7) family Kv channels (M-channels)[2, 3], is the predominant Kv current regulating neuronal excitability[4]. KCNQ channels, like all other known Kv channels, form as tetramers of α subunits, each subunit of which contains six transmembrane (S) segments, split into the voltage-sensing domain (VSD, S1–4) and the pore module (S5–6). The M-current is generated primarily by KCNQ2 and KCNQ3 subunits, most commonly in the form of heteromeric (KCNQ2/3) channels[4], but homomers also exist in vivo and KCNQ4 and KCNQ5 also contribute[5]; KCNQ5, for example, generates hippocampal afterhyperpolarization currents[6]. Because KCNQ2/3 channels are receptor-regulated, non-inactivating, and open at relatively hyperpolarized membrane potentials, they are uniquely suited to regulate neuronal excitability, and their gating state dictates phasic versus tonic firing of neurons[4].

KCNQ2 subunits are particularly enriched in GABAergic neurons, especially those important for control of network oscillations and synchrony[7]. Presynaptic KCNQ2/3 channels are thought to modulate glutamate and γ-aminobutyric acid (GABA) release, and are suggested to act pre- and post-synaptically to suppress neuronal excitability[8, 9], and Kcnq2-deficient mice exhibit abnormal GABAergic synaptic transmission[10].

KCNQ2–5 are therapeutically activated by a class of anticonvulsants typified by retigabine and ML-213, which facilitate channel opening at more negative membrane potentials. KCNQ3 is the most sensitive to retigabine[11], although KCNQ2–5 all contain a transmembrane segment 5 (S5) tryptophan (W265 in human KCNQ3, W270 in KCNQ5) that is essential for sensitivity to this class of anticonvulsants[12], yet has no known physiological role. Here, phylogenetic analysis revealed that W265/270 is highly conserved, for >500 million years in deuterostomes, and is also present in some Cnidaria, e.g., Hydrozoans. In contrast, W265/270 was absent from all protostome KCNQs we analyzed (Fig. 1a–c). Given this enduring conservation, we hypothesized that W265/270 evolved as a chemosensor for native ligands present in both primitive and modern nervous systems. Here, we show that KCNQ3-W265 forms part of a high-affinity binding site for the neurotransmitter GABA and related metabolites, binding of which potentiates channel activation especially at subthreshold potentials, resulting in membrane hyperpolarization. This first description of direct activation of voltage-dependent ion channels by a neurotransmitter uncovers a novel mechanism for controlling cellular excitability.

## Results

### KCNQ2-5 channels contain GABA binding sites. 
Focusing on chemical properties required for anticonvulsant action of retigabine and related compounds on KCNQ2/3, including negative electrostatic surface potential localized close to a carbonyl oxygen[13], we identified GABA as a candidate. GABA, the primary inhibitory neurotransmitter, bears strong negative electrostatic surface potential near its carbonyl oxygen, in contrast to the chemically related excitatory neurotransmitter, glutamate (Fig. 1d). Strikingly, in silico docking predicted binding of GABA to KCNQ3-W265, similar to retigabine and ML-213 (Fig. 1e). GABA does not readily cross the plasma membrane, but W265 lies in a crevice apparently accessible from the extracellular side (Fig. 1f); this portion of S5 also appeared accessible in the related Kv1.2–Kv2.1 "paddle-chimera" structure, which includes lipids[14]

(Fig. 1g). Accordingly, Xenopus oocyte expressed homomeric wild-type human KCNQ2–5-bound extracellularly applied ³H-GABA, while KCNQ1, which lacks W265, did not (Fig. 1h; Supplementary Fig. 1). Further indicative of the specificity of ³H-GABA binding to KCNQ2–5, non-injected oocytes bound minimal ³H-GABA (Supplementary Fig. 1) and ³H-GABA binding to homomeric wild-type human KCNQ3 was outcompeted by co-incubation with cold (unlabeled) GABA (1 mM) (Fig. 1i). Finally, results of saturation binding experiments conducted with oocyte-expressed homomeric wild-type human KCNQ3 using 10 different concentrations of ³H-GABA were consistent with binding to a specific site in KCNQ3 with a $K_d$ of 126 nM and a $B_{max}$ of 27 fmol per oocyte (Fig. 1j; see Supplementary Fig. 1d for an expanded view of the low [GABA] portion of the curve). The $B_{max}$ value was similar to results obtained from previous intact oocyte radioligand binding studies of other heterologously expressed membrane proteins[15].

### GABA activates KCNQ3 and KCNQ5. 
Using oocyte expression and two-electrode voltage clamp (TEVC), we found that GABA activates KCNQ3* (an expression-optimized KCNQ3-A315T mutant that ensures robust currents[16]) and KCNQ5, especially at subthreshold potentials. In contrast, the activity of KCNQ1, 2 and 4 was relatively GABA-insensitive (Fig. 2a; Supplementary Figures 2–6; Supplementary Tables 1–5). Thus, the KCNQ3-W265 equivalent is important for GABA binding (Fig. 1) but not sufficient for KCNQ activation by GABA (Fig. 2a). The KCNQ3* GABA $EC_{50}$ was $1.0 \pm 0.1$ µM at −60 mV (Supplementary Table 9), a higher sensitivity than that previously reported for retigabine (11.6 µM) or ML-213 (3.6 µM)[13]. This was a lower potency than the $K_d$ we observed for GABA binding to wild-type KCNQ3 (126 nM; Fig. 1j); but see GABA effects on KCNQ2/3-dependent membrane hyperpolarization, below (Fig. 2h). GABA (10 µM) shifted the $V_{0.5}$ of activation of KCNQ3* by −8 mV, but this understates the efficacy of GABA in the activation of KCNQ3*, because there is also a general increase in channel activity across the voltage range from −60 mV to +20 mV (Fig. 2a, b; Supplementary Fig. 4). The KCNQ5 GABA $EC_{50}$ was $0.06 \pm 0.01$ µM at −60 mV, a 17-fold higher GABA sensitivity than for KCNQ3* (Supplementary Table 9), but the efficacy of GABA with respect to channel activation was more than threefold greater for KCNQ3* compared to KCNQ5 (Fig. 2a, b). GABA (10 µM) shifted the KCNQ5 $V_{0.5}$ of activation by −15 mV (Supplementary Figure 6; Supplementary Table 5).

### GABA activation of KCNQ2/3 channels. 
We next focused on heteromeric KCNQ2/3 channels, the primary molecular correlate of neuronal M-current[4]. GABA activated KCNQ2/3 with increasing efficacy at more negative membrane potentials, a property important for subduing neuronal activity; thus GABA (10 µM) negative-shifted the $V_{0.5}$ of KCNQ2/3 activation by −14 mV (Fig. 2c–e; Supplementary Fig. 7; Supplementary Table 6). At −60 mV, 10 µM GABA increased KCNQ2/3 current fourfold (Fig. 2f). The $EC_{50}$ for KCNQ2/3 activation was $0.85 \pm 0.1$ µM GABA at −60 mV (Supplementary Table 9), while the excitatory neurotransmitter glutamate had no effect on KCNQ2/3 even at 10 mM (Fig. 2g; Supplementary Fig. 8; Supplementary Table 7). Reflecting its greater efficacy at more negative membrane potentials and the efficiency of KCNQ channels in setting membrane potential ($E_M$), GABA exerted potent effects on KCNQ2/3-dependent membrane hyperpolarization ($EC_{50}$, 120 nM) (Fig. 2h). Canonical pentameric GABA_A receptors, which are triggered to open when GABA binds to an extracellularly located, inter-subunit GABA-binding site, participate in either phasic or tonic inhibition, depending on their location and

subunit composition[17, 18]. Tonic extracellular GABA concentration in mammalian brain is calculated to be ~160 nM[19, 20], and transient peak GABA concentrations of several millimolar occur in the synaptic cleft[21]. Canonical $\alpha_x\beta_3\gamma_2$ GABA$_A$Rs exhibit GABA EC$_{50}$ values of 1.0–157 μM[22, 23]. Thus, KCNQ2/3 channel GABA affinity compares to that of the most sensitive $\alpha_x\beta_3\gamma_2$ GABA$_A$Rs. GABA begins to activate KCNQ2/3 immediately on wash-in; the current augmentation takes ~3 min to plateau, and persists

during wash-out (Fig. 2i). Time to onset of functional effects upon wash-in did not appear voltage-dependent, as it was not altered by switching to +60 mV during wash-in; the GABA-augmented current was readily inhibited by washing in the KCNQ channel inhibitor, XE991 (50 μM) (Fig. 2j). GABA accelerated KCNQ2/3 activation and slowed deactivation, suggesting GABA stabilizes the KCNQ2/3 open state (Fig. 2k; Supplementary Fig. 9; Supplementary Table 8).

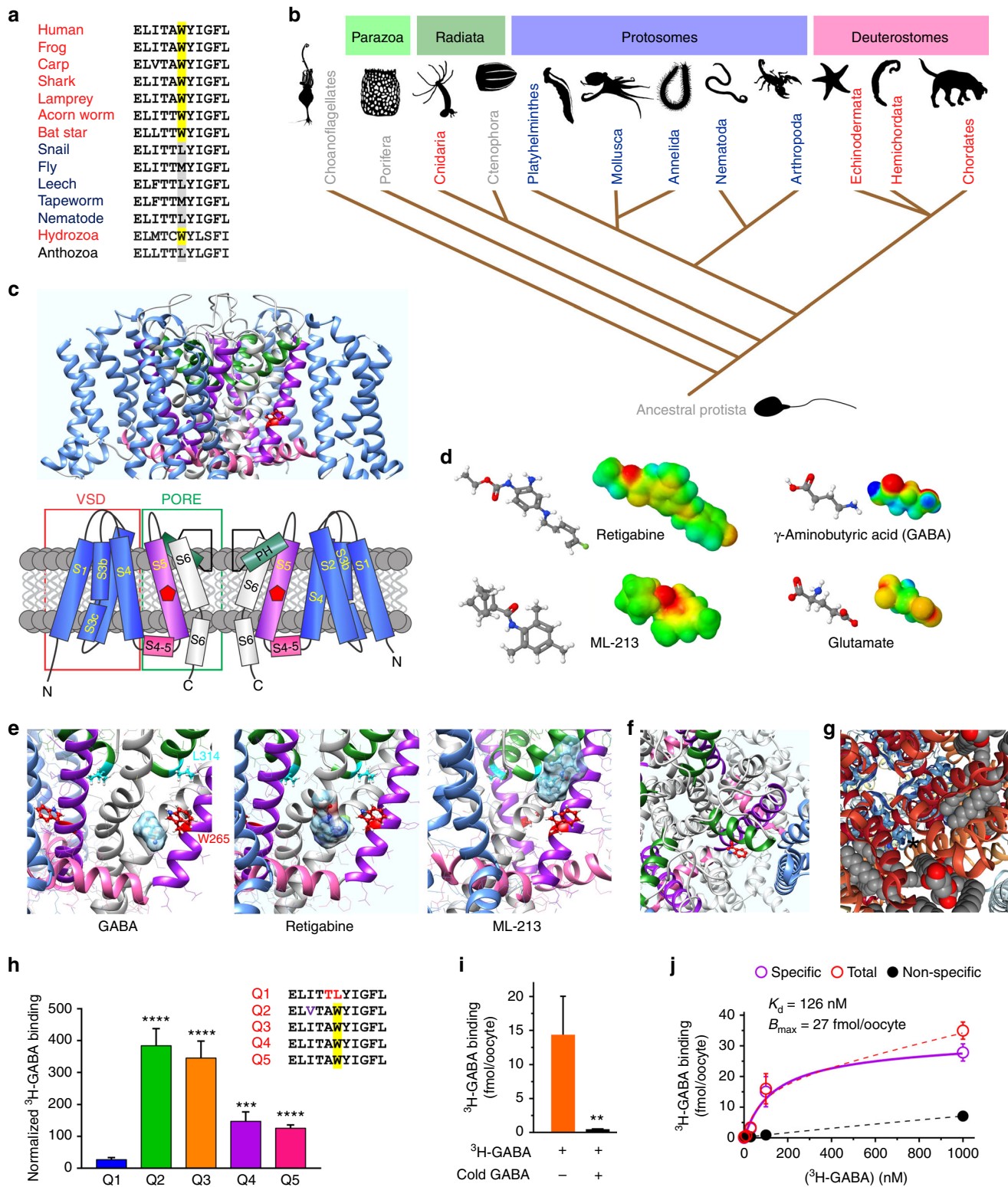

**GABA activates KCNQ2/3 channels by binding to KCNQ3-W265**. Our radioligand binding data (Fig. 1h–j) demonstrate direct binding of GABA to KCNQ2–5, but we nevertheless tested for other possibilities and artifacts. At 100 μM, GABA neither induced current in, nor shifted the resting membrane potential of, water-injected oocytes, demonstrating our observations did not arise from an endogenous ion channel (Fig. 3a), neither did increasing GABA to 1 mM produce any noticeable effects (Supplementary Fig. 10). Neither did GABA, even at 1 mM, affect activity of KCNA1 (Kv1.1), a Kv channel from a different subfamily (Fig. 3b; Supplementary Fig. 11; Supplementary Table 10). GABA$_B$ receptors are metabotropic receptors that can activate some potassium channels via pertussis-sensitive G proteins[24]. In previous studies, it was concluded that *Xenopus* oocytes do not express endogenous GABA$_B$ receptors[25], which can activate some potassium channel subtypes (not reported for KCNQs), but we nevertheless tested for their possible role in the observed effects, using four approaches. These studies ruled out a role for GABA$_B$ receptors in KCNQ activation by GABA, as follows. First, GABA activated KCNQ2/3 even after overnight treatment of oocytes with pertussis toxin, which inhibits G proteins associated with GABA$_B$ activation of some potassium channels (Fig. 3c; Supplementary Fig. 12; Supplementary Table 11). Second, KCNQ2/3 channels were insensitive to baclofen (100 μM), a GABA analog that acts as a GABA$_B$R agonist[26] (Fig. 3d; Supplementary Fig. 13; Supplementary Table 12). Third, GABA$_B$ receptor antagonist saclofen neither altered KCNQ2/3 currents alone, nor did it diminish GABA activation of KCNQ2/3 (Fig. 3e; Supplementary Figs. 14, 15; Supplementary Tables 13, 14). Fourth, the potent GABA$_B$ receptor antagonist CGP55845 neither altered KCNQ2/3 currents alone, nor did it diminish GABA activation of KCNQ2/3 (Fig. 3f; Supplementary Figs. 16, 17; Supplementary Tables 15, 16).

Mutagenesis studies further cemented the conclusion that GABA directly modulates KCNQ2/3 channels. Reflecting our in silico predictions, $^3$H-GABA binding results (Fig. 1e–j), and KCNQ2 GABA-insensitivity (Fig. 2a), a mutant KCNQ2/3 channel complex in which the essential KCNQ3 tryptophan was replaced with leucine (KCNQ2/KCNQ3-W265L) was GABA-insensitive (up to 10 mM), as was double tryptophan-mutant heteromer KCNQ2-W236L/KCNQ3-W265L (Fig. 3g–i; Supplementary Figs. 18, 19; Supplementary Table 17), which also showed impaired $^3$H-GABA binding (Fig. 3j).

**GABA activates endogenous neuronal M-channels**. Switching to mammalian cells, KCNQ2/3 channels heterologously expressed in Chinese hamster ovary (CHO) cells were likewise activated by GABA, demonstrating this was not an oocyte-specific phenomenon (Fig. 4a, b). GABA (100 μM) shifted the midpoint voltage dependence of KCNQ2/3 activation by −14.6 mV (from −2.9 to −17.5 mV, Supplementary Table 18), similar to the shift we observed in oocytes (−14 mV; Fig. 2c and Supplementary Table 6).

We next tested the effected of GABA on native M-current in PC12 cells, a rat pheochromocytoma cell line of neural crest origin that retains many of the properties of dopaminergic neurons and has previously been used to study M-current[27, 28]. Culturing PC12 cells for 4–7 days with NGF resulted in extensive neurite outgrowth and increased expression of native voltage-gated potassium currents (Fig. 4c), as previously reported[27, 28]. We found that 8/10 PC12 cells studied expressed potassium current sensitive to the KCNQ channel blocker XE991 (10 μM). Importantly, in seven of these eight cells, GABA (100 μM) negative-shifted the voltage dependence of the PC12 cell tail current activation, in the presence of picrotoxin and CGP55845 to block native canonical GABA$_A$ and GABA$_B$ receptors, respectively. The mean shift of −13.5 mV in the midpoint of voltage-dependent activation in PC12 cells in response to 100 μM GABA, in the presence of canonical GABAR blockers (Supplementary Table 19), was similar to that for KCNQ2/3 channels expressed in oocytes and in CHO cells (−14 mV) with 100 μM GABA application. Application of KCNQ channel blocker XE991 (10 μM) removed the GABA-dependent negative-shift in the PC12 cell tail currents (Fig. 4d).

We also recorded the effects of GABA on dorsal root ganglion (DRG) neurons isolated from adult mice, using perforated patch-clamp whole-cell recordings. Cells were held at −20 mV to inactivate other Kv channels and then pulsed to −60 mV, after wash-in of inhibitors of canonical GABA$_A$ and GABA$_B$ receptors (100–300 μM picrotoxin; 10 μM CGP55845, respectively), first in the absence, and then in the presence, of GABA (100 μM). XE991 (10 μM) was subsequently washed in, in the presence of the other drugs, to inhibit M-current. From a holding potential of −20 mV, pulses to −60 mV produced a partial deactivation and then stabilization characteristic of M-current; current at −60 mV was increased by GABA and this current was decreased by XE991 (Fig. 4e). GABA (100 μM) increased the relative proportion of −60 mV current to that of −20 mV current, consistent with greater GABA efficacy at more negative voltages (Fig. 4f). Accordingly, the mean GABA-dependent current increase was threefold at −60 mV, but there was no mean change in current at −20 mV (Fig. 4g). We also quantified the deactivating portion of the −60 mV current in isolation. GABA decreased this deactivating current by ~25%, consistent with fewer channels closing upon the transition from −20 mV to −60 mV, in the presence of

---

**Fig. 1** Neuronal KCNQs contain an evolutionarily conserved GABA-binding pocket. **a** Sequence alignment of S5 residues 265–275 (human numbering) in KCNQ5 or the closest equivalent in representative organisms from the clades shown in **b**. Yellow, tryptophan essential for drug binding (KCNQ5-W270; KCNQ3-W265). Organism descriptors: black, single KCNQ gene in genome, no W270 equivalent; dark blue, multiple *KCNQ* genes, no W270 equivalent; red, multiple KCNQ genes, at least one containing W270 equivalent. **b** Phylogeny (distances not intended to equate to time) indicating evolutionary relationships from ancestral Protista to Chordates and other major clades. Text colors as in **a**; gray, no *KCNQ* genes. **c** Upper, chimeric KCNQ1/KCNQ3 structural model (red, KCNQ3-W265; cyan, KCNQ3-L314); lower, topological representation (pentagon, KCNQ3-W265). **d** Electrostatic surface potentials (red, electron-dense; blue, electron-poor; green, neutral) and structures calculated and plotted using Jmol. **e** Close-up side views of KCNQ structure as in **c**, showing results of SwissDock unguided in silico docking of GABA, retigabine, and ML-213. **f** View from extracellular side of KCNQ structure showing W265 accessibility. **g** View from extracellular side of Kv1.2-Kv2.1 "paddle-chimera" structure showing S5 accessibility (*) despite lipid presence (lipid spacefills shown). **h** Mean $^3$H-GABA binding, normalized to channel surface expression, for KCNQ channels expressed in *Xenopus* oocytes; $n = 47$ (Q1), 14 (Q2), 48 (Q3), 30 (Q4), 44 (Q5) in two to four batches of oocytes; ***$P = 0.0003$; ****$P < 0.0001$; versus KCNQ1. Inset, sequence alignment of S5 265–275 (KCNQ5 numbering) for human KCNQ1–5. Yellow, KCNQ5-W270. Error bars indicate SEM. **i** Mean $^3$H-GABA binding to KCNQ3 expressed in *Xenopus* oocytes, with (+) versus without (−) competition from unlabeled (cold) GABA (1 mM); $n = 20$ per group; **$P = 0.007$. Error bars indicate SEM. **j** Saturation binding studies using $^3$H-GABA and oocyte-expressed KCNQ3. Ten different $^3$H-GABA concentrations were used (0.1–1000 nM; $n = 20$ oocytes per group). Mean non-specific binding was quantified in parallel experiments similar except for the addition of 1 mM cold GABA. Mean specific binding was calculated by subtracting mean non-specific from total binding. Error bars indicate SEM

GABA. XE991 applied in combination with GABA inhibited this deactivating current >50%, consistent with the majority of the current being passed by KCNQ channels (Fig. 4h). Thus, GABA activation of M-channels is direct, independent of the expression system, and occurs with both exogenously and endogenously expressed channels.

**GABA overcomes muscarinic inhibition of KCNQ2/3 channels.** Muscarinic inhibition of KCNQ channels is achieved by disruption of KCNQ regulation by phosphatidylinositol 4,5-bisphosphate ($PIP_2$). $PIP_2$ facilitates KCNQ channel opening, thus

hyperpolarizing KCNQ channel voltage-dependent activation, enabling opening at more negative membrane potentials. Muscarinic acetylcholine receptor (mAChR) activation disrupts this, by hydrolyzing $PIP_2$ and/or by reducing $PIP_2$ sensitivity, thus positive-shifting KCNQ channel voltage dependence of activation, inhibiting activity particularly at negative potentials[29]. The lack of GABA responsiveness of KCNQ1, 2, and 4 (Fig. 2a) effectively ruled out the possibility that endogenous mAChRs/ changes in $PIP_2$ were mediating GABA effects on KCNQ3. However, we further tested this and also asked whether GABA could overcome muscarinic inhibition of KCNQ2/3. For these

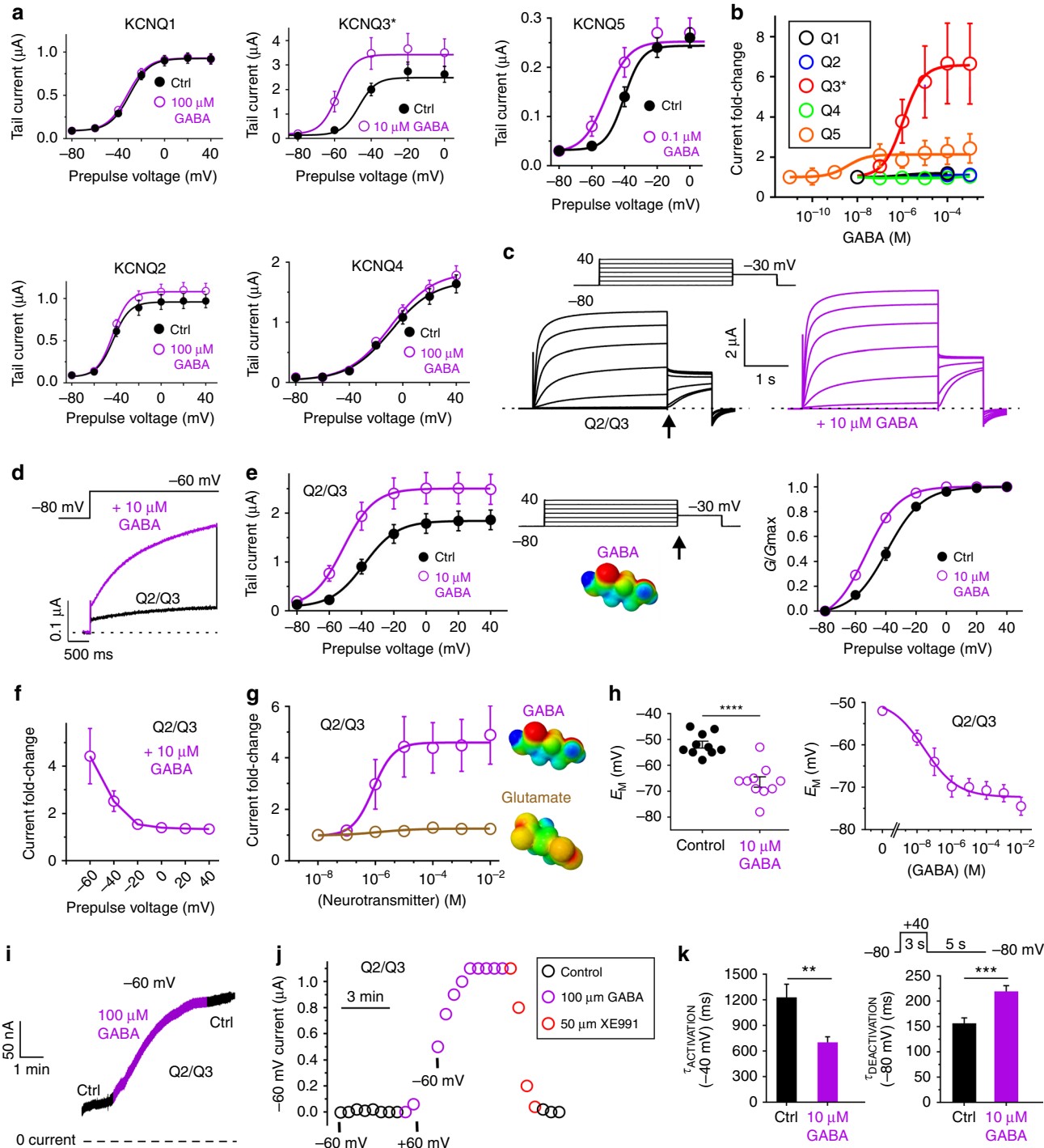

experiments, we reverted to the *Xenopus* oocyte expression system. As expected, application of acetylcholine (ACh) to activate endogenous oocyte mAChRs inhibited KCNQ2/3 activity, especially at negative voltages; maximal effect occurred at 1 μM ACh and higher (Fig. 5a, b; Supplementary Fig. 20; Supplementary Table 20). For comparison, application of a 100-fold greater concentration of dopamine (100 μM) had no effect on KCNQ2/3 current (Fig. 5c; Supplementary Fig. 21; Supplementary Table 21). ACh inhibition of KCNQ2/3 was prevented by the mAChR antagonist atropine, indicating that ACh inhibited KCNQ2/3 solely by the canonical, indirect pathway requiring mAChR activation (Fig. 5b, d; Supplementary Fig. 22). Atropine itself had no direct effect on KCNQ2/3 activity (Fig. 5b, e; Supplementary Fig. 23; Supplementary Table 22). Additionally, atropine did not inhibit GABA activation of KCNQ2/3 (Fig. 5f; Supplementary Fig. 24; Supplementary Table 23), ruling out an indirect, mAChR-dependent mechanism for GABA activation of KCNQ2/3, as expected from the GABA-insensitivity of other $PIP_2$-sensitive KCNQs (Fig. 2a).

Importantly, GABA overcame ACh-dependent KCNQ2/3 inhibition (Fig. 5g; Supplementary Fig. 25; Supplementary Table 24), indicating that GABA can override the canonical muscarinic inhibition of KCNQ2/3 for which the M-current is named. To further test this conclusion, we pre-incubated KCNQ2/3-expressing oocytes with the phosphoinositide 3/4-kinase inhibitor, wortmannin (30 μM for 3 h), to prevent $PIP_2$ synthesis and partially deplete intracellular $PIP_2$. Under these conditions, KCNQ2/3 was still robustly activated by GABA, which now (at 10 μM) shifted the $V_{0.5}$ of KCNQ2/3 activation by −24 mV (Fig. 5h; Supplementary Fig. 26; Supplementary Table 25). Thus, GABA activation of KCNQ2/3 can overcome the inhibitory effects of mAChR activation or direct $PIP_2$ depletion/prevention of $PIP_2$ synthesis. As a final control for the independence of GABA from mAChR regulation, we examined effects of GABA on KCNQ1-KCNE1, a heteromeric channel with 100-fold higher $PIP_2$ sensitivity than homomeric KCNQ1[30]. We found no effect of GABA on KCNQ1-KCNE1 activity, nor on the resting membrane potential of KCNQ1-KCNE1-expressing oocytes (Fig. 5i, j; Supplementary Fig. 27; Supplementary Table 26). Thus, KCNQ activation by GABA does not involve mAChRs or $PIP_2$, and indeed can overcome these factors in KCNQ2/3 channels.

**GABA analogs compete for KCNQ3-W265 binding**. GABA coexists in the CNS with various analogs and metabolites, two of which, β-hydroxybutyrate (BHB) and γ-amino-β-hydroxybutyrate (GABOB), we discovered to also activate KCNQ2/3. GABOB had a higher affinity (EC$_{50}$ of 0.12 μM), but

was a lower efficacy partial agonist, for KCNQ2/3 compared to BHB (EC$_{50}$ of 0.94 μM) and to GABA (EC$_{50}$ of 0.85 μM). In contrast, the structurally related straight-chain alkyl carboxylic acid, valeric acid, was inactive up to 5 mM (Fig. 6a, b; Supplementary Figs. 28–30; Supplementary Tables 27–29). The high sensitivity of KCNQ2/3 for BHB is in contrast with that of canonical GABA$_A$ receptors, which are BHB-insensitive up to 5 mM BHB[31]. In silico docking studies suggested that, like GABA, BHB, and GABOB bind to KCNQ3-W265 (Fig. 6c). Accordingly, GABOB, a higher-affinity partial agonist compared to GABA, diminished the KCNQ2/3-activating efficacy of GABA (Fig. 6d; Supplementary Fig. 31; Supplementary Table 30). In contrast and as expected, glutamate, which we found to be inactive on KCNQ2/3 (Fig. 2g), did not alter GABA activation of KCNQ2/3 (Fig. 6e; Supplementary Fig. 32; Supplementary Table 31). Finally, we found that GABOB reduces the efficacy of retigabine, an anticonvulsant that also activates KCNQ2/3 by binding to the W265 equivalent (Fig. 6f).

**Discussion**

Prior phylogenetic analyses of KCNQ sequences showed their absence in Porifera (sponges) and ctenophores, and suggested that KCNQ emerged as a single gene in a cnidarian/bilaterian ancestor. The KCNQ encoded by this single gene evolved $PIP_2$ sensitivity even before the Bilaterian/Cnidarian divergence, indicating the importance of this feature as early as 600 million years ago[32]. This $PIP_2$ sensitivity developed before the duplication of KCNQ into genes orthologous to present-day KCNQ1 and KCNQ5, creating multiple KCNQ genes per genome. Some cnidarians, including *Hydra* (a Hydrozoan) have multiple *KCNQ* genes, whereas others, such as *Nematostella* (an Anthozoan) have only one. Previous phylogenetic analysis suggested that cnidarian KCNQs do not fit into either of the KCNQ clades present in bilaterian genomes, which are split into KCNQ1-like and KCNQ2-like[32]. As protostome KCNQs lack the W270 equivalent (Fig. 1a, b), it is interesting to ponder whether cnidarian KCNQs evolved the S5 W independently to the deuterostomes, or whether protostomes lost a W already present in a cnidarian/bilaterian ancestor. Either way, this places emergence of KCNQ5-W270 or its equivalent much earlier than evolution of the anchor motif for interaction with Ankyrin G, required for location at axon initial segments, which appeared in KCNQs in the time period of divergence of extant jawless and jawed vertebrates[33]. For comparison, *Hydra* are also known to express functional canonical GABA$_A$ receptors with pharmacology similar in some respect to mammalian GABA$_A$ receptors[34]. GABA itself is found in all organisms; glutamate decarboxylase (GAD), which catalyzes conversion of glutamate into GABA, is found in bacteria (where

**Fig. 2** KCNQ3 and KCNQ5 are activated by GABA in *Xenopus* oocytes. All error bars in figure indicate SEM. **a** Mean tail current versus prepulse voltage relationships recorded by TEVC in *Xenopus laevis* oocytes expressing homomeric KCNQ1–5 channels in the absence (black) and presence (purple) of GABA ($n = 4$–8). Voltage protocol as in **c**. **b** Mean GABA dose response at −60 mV for KCNQ1–5, quantified from data as in **a** ($n = 5$–8). **c** TEVC recordings using a current-voltage family protocol (upper left inset) in oocytes expressing KCNQ2/3 channels in the absence (black) and presence (purple) of GABA (10 μM). Dashed line indicates zero current level in this and all following current traces. **d** Averaged current traces at −60 mV for KCNQ2/3 channels in the absence (black) and presence (purple) of GABA (10 μM) ($n = 10$). Voltage protocol upper inset. **e** Left, mean tail current; right, mean normalized tail current ($G/G$max), both measured at arrow in voltage protocol (center) from traces as in **c**, with versus without GABA (10 μM) ($n = 10$). **f** Mean voltage dependence of KCNQ2/3 current fold-increase by GABA (10 μM), plotted from traces as in **c** ($n = 10$). **g** Mean dose response of KCNQ2/3 channels at −60 mV for GABA (calculated EC$_{50} = 0.85$ μM; $n = 10$) and glutamate (no effect; $n = 5$). **h** GABA hyperpolarizes resting membrane potential ($E_M$) of unclamped oocytes expressing KCNQ2/3. Left, effects of 10 μM GABA; right, dose response; $n = 10$, ****$P < 0.0001$. **i** Exemplar −60 mV KCNQ2/3 current before (left, black), during wash-in of GABA (purple) and after wash-out (right, black). **j** Exemplar −60 mV KCNQ2/3 current before (left, black), during wash-in of GABA (purple), wash-in of XE991 (red), and after wash-out (right, black). Membrane potential was clamped at −60 mV except for a 2-min pulse to +60 mV during the early phase of GABA wash-in. **k** Mean activation (left) and deactivation (right) rates for KCNQ2/3 before (control) and after wash-in of GABA ($n = 10$); **$P < 0.01$; ***$P = 0.0007$. Activation rate was quantified using voltage protocol as in **c**. Deactivation rate was quantified using voltage protocol shown above

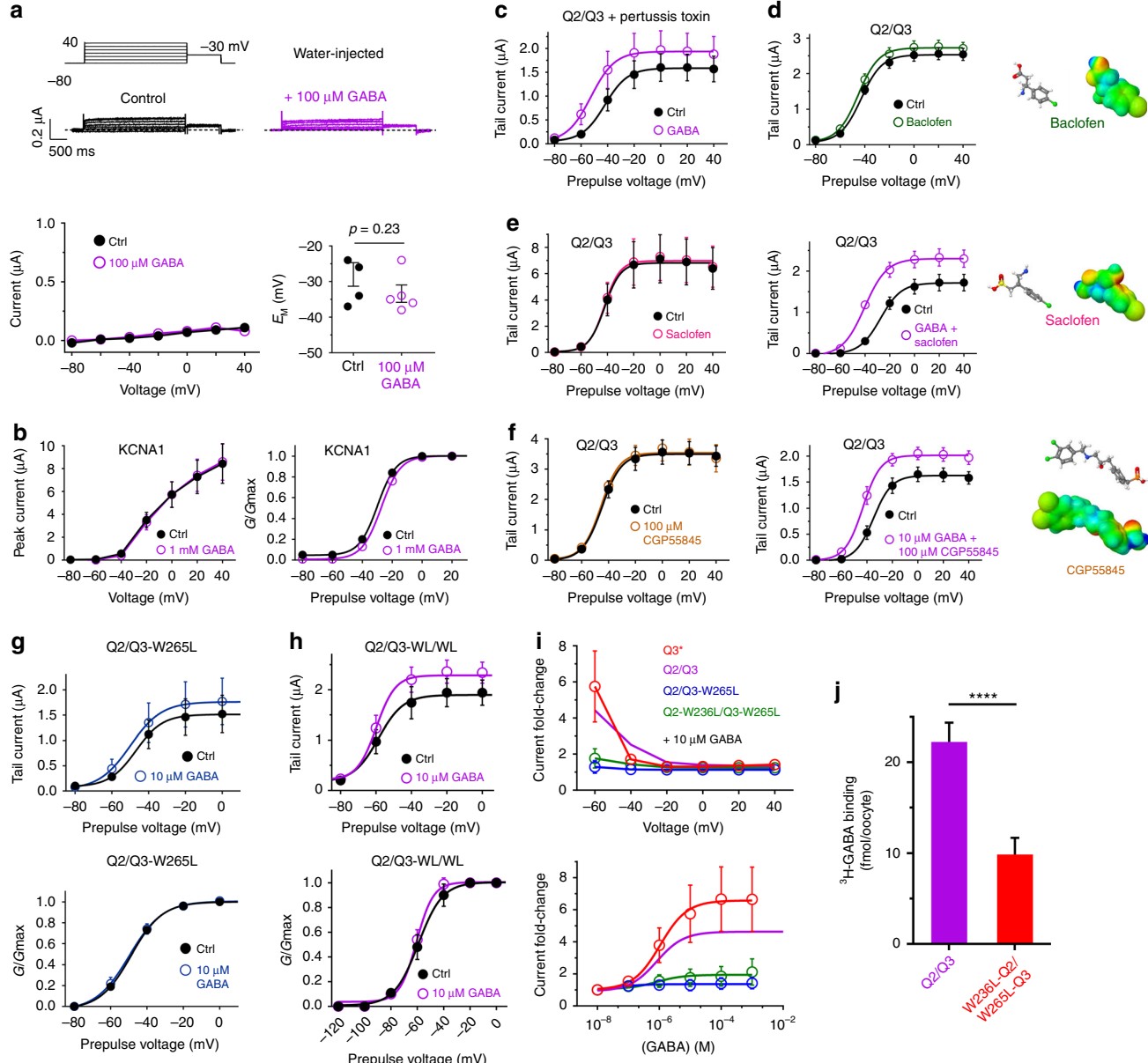

**Fig. 3** GABA directly opens KCNQ2/3 channels. All error bars in figure indicate SEM. **a** TEVC of water-injected *Xenopus laevis* oocytes showing no effect of GABA (100 μM) on endogenous currents or (lower right) resting membrane potential ($E_M$) ($n = 4$–5). **b** TEVC of *Xenopus laevis* oocytes expressing KCNA1 (Kv1.1) showing no effect of GABA (1 mM) on peak current (left) or normalized tail current (right) ($n = 4$). **c**–**f** GABA effects on KCNQ2/3 in oocytes do not require GABA$_B$ receptor activity. Mean KCNQ2/3 tail currents in oocytes showing GABA activates KCNQ2/3 in the presence of pertussis toxin (2 μg/ ml), saclofen (100 μM), or CGP55845 (100 μM) and that neither baclofen (100 μM), saclofen, nor CGP55845 alter KCNQ2/3 current independently ($n = $ 5–7). **g** Upper, tail current; lower, normalized conductance; showing mean GABA response of oocyte-expressed KCNQ2/KCNQ3-W265L ($n = 5$). **h** Upper, tail current; lower, normalized conductance; showing mean GABA response of oocyte-expressed KCNQ2-W236L/KCNQ3-W265 (Q2/Q3-WL/WL) ($n = $ 5). **i** Mean current fold-changes (upper) and dose responses (lower) for channels as indicated; KCNQ2/KCNQ3 results (purple line) and KCNQ3* results (red line) from Fig. 2 shown for comparison; $n = 5$–10. **j** Mean ³H-GABA binding for KCNQ2/3 versus KCNQ2-W236L/KCNQ3-W265L channels expressed in *Xenopus* oocytes; $n = 73$ (Q2/Q3), 29 (KCNQ2-W236L/KCNQ3-W265L) in two to four batches of oocytes; ****$P < 0.0001$

decarboxylation of amino acids including glutamate aids in acid resistance[35]) and plants (e.g., in soybean[36]) in addition to animals.

KCNQ5-W270/KCNQ3-W265 are necessary for binding but not sufficient for GABA-dependent activation; KCNQ2 and KCNQ4 each bear an equivalent residue and bind GABA but are not activated by GABA, analogous to but more extreme than the lower retigabine sensitivity of KCNQ2 versus KCNQ3[37]. Thus, other residues are important, for determining binding affinity/ precise position and/or communicating GABA binding to the

KCNQ gating machinery. The finding that KCNQ5, the modern KCNQ most closely related to the progenitor KCNQ found in primitive deuterostome nervous systems, is GABA-sensitive, is consistent with the idea that the ability to activate in response to GABA was a positive selection pressure contributing to conservation of KCNQ5-W270.

This also suggests a plausible explanation for why modern, neuronal KCNQs bear the W265 equivalent while the modern-day cardiac and epithelial KCNQ1 subunit does not. After the founder KCNQ gene was duplicated in early bilaterians, tens of

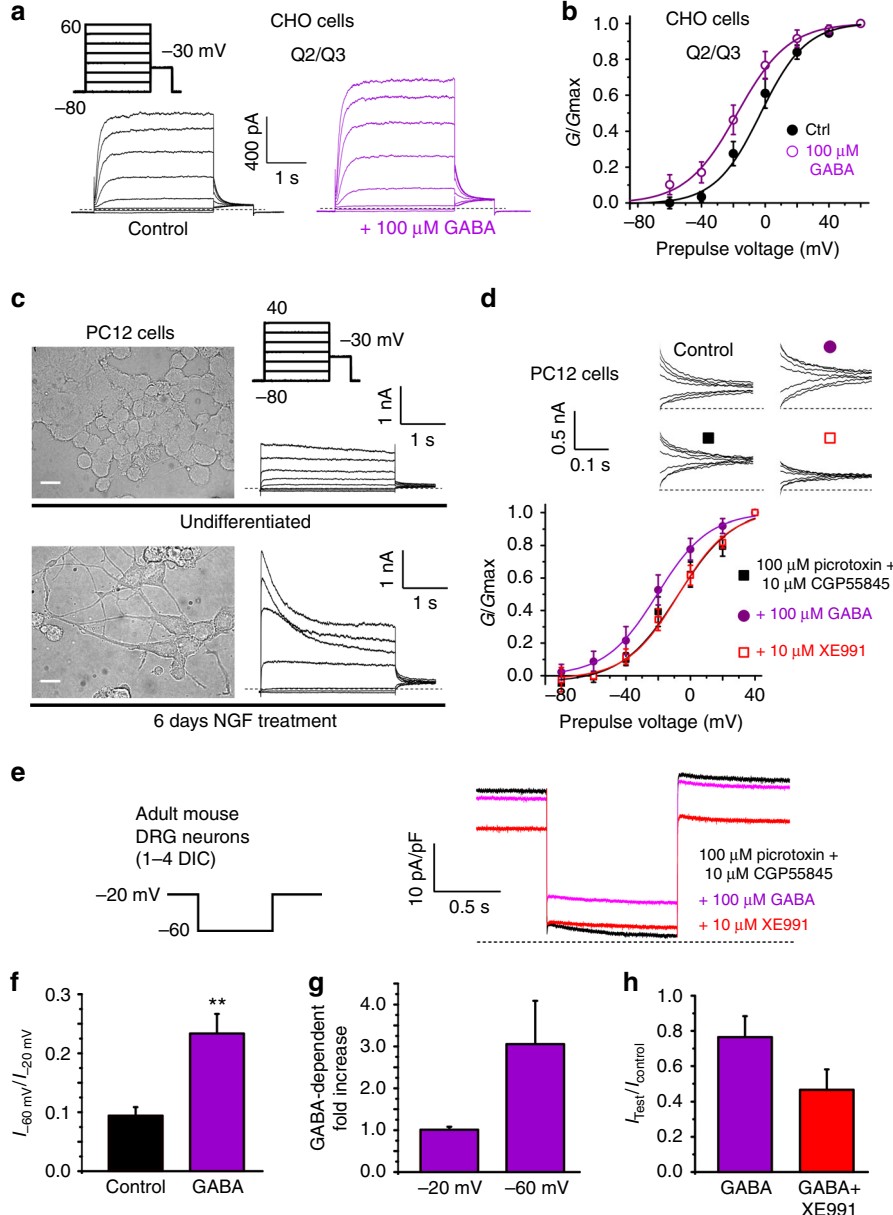

**Fig. 4** GABA activates KCNQ2/3 in CHO cells and native M-current. All error bars in figure indicate SEM. **a** Exemplar current traces from CHO cells transfected to express KCNQ2/3 channels, recorded using whole-cell patch clamp, showing effects of 100 µM GABA ($n = 6$). **b** Mean normalized tail current from recordings as in **a** ($n = 6$). **c** Representative micrographs (left) and whole-cell currents (right) from undifferentiated (upper) versus nerve growth factor (NGF)-differentiated (lower) PC12 cells. Scale bars, 10 µm. **d** Upper, representative tail currents (using voltage protocol as in **c**); lower, mean normalized tail currents, recorded from NGF-differentiated PC12 cells bathed in extracellular solution alone (control) or with picrotoxin (100 µM) and CGP55845 (10 µM) to block GABA_A and GABA_B receptors, respectively, alone (black square) or in combination with GABA (100 µM; purple circle) or XE991 (10 µM; open red square); $n = 7$. **e** Representative tail currents (using voltage protocol on left) recorded from mouse DRG neurons bathed in extracellular solution containing picrotoxin (100 µM) and CGP55845 (10 µM), alone (black) or in combination with GABA (100 µM; purple) or GABA + XE991 (10 µM; red). **f** Mean current at −60 mV divided by current at −20 mV in the same DRG neuron, using the protocol as in **e**, in the absence (control) or presence of GABA ($n = 7$); **$P = 0.005$. **g** Mean GABA-dependent increase in current at −20 mV versus at −60 mV in DRG neurons, using the protocol as in **e** ($n = 7$); $P = 0.07$ between groups. **h** Mean effect of GABA versus GABA + XE991 on the magnitude of the deactivating current at −60 mV, using the protocol as in **e** ($n = 5$); $P = 0.1$ between groups

millions of years later when nervous systems were centralizing in deuterostomes, there was a positive selective pressure for the progenitor neuronal KCNQ (most similar to mammalian KCNQ5), but not non-neuronal KCNQ1, to sense GABA, and KCNQ5-W270 or its equivalent emerged and is conserved in deuterostomes to the present day. When the other neuronal KCNQs (KCNQ2, 3 and 4) emerged, they retained the ability to

bind GABA but only one (KCNQ3) is also activated by GABA. The functional complexity in terms of responsiveness to GABA may help explain why several different KCNQs are present in modern nervous systems, as the various homomeric and heteromeric KCNQ channels can each respond to GABA differently. The evolution of KCNQ GABA sensitivity also paved the way for cardiac-safe KCNQ2–5 openers, which require the tryptophan

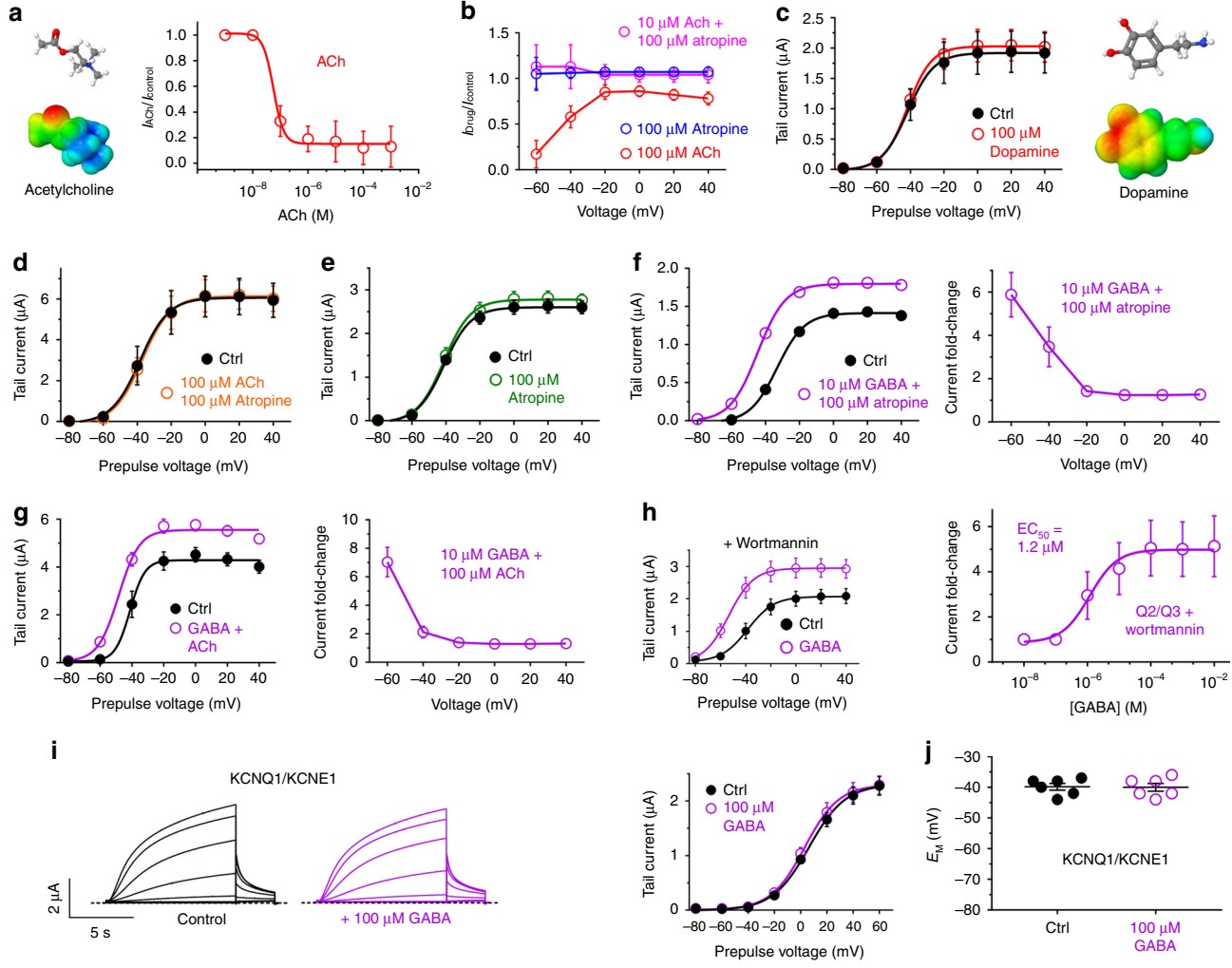

**Fig. 5** GABA activation is independent of and overrides muscarinic inhibition of KCNQ2/3 in *Xenopus* oocytes. All error bars in figure indicate SEM. **a** Mean dose response for acetylcholine (ACh) on KCNQ2/3 activity in oocytes ($n = 5$). **b** Mean effects versus voltage for ACh or atropine alone or in combination on KCNQ2/3 activity in oocytes ($n = 5$). **c** Lack of effects of dopamine (100 μM) on KCNQ2/3 mean tail currents in oocytes ($n = 6$). **d** Atropine blocks KCNQ2/3 inhibition by ACh (measured using tail currents in oocytes; $n = 5$). **e** Lack of effects of atropine (100 μM) on KCNQ2/3 mean tail currents in oocytes ($n = 5$). **f** Atropine does not prevent GABA activation of KCNQ2/3 measured via tail currents in oocytes ($n = 5$). Left, mean tail currents, right, current fold-change versus voltage for GABA + atropine versus no drugs. **g** GABA prevents inhibition of KCNQ2/3 by ACh, measured via tail currents in oocytes ($n = 5$). Left, mean tail currents, right, current fold-change versus voltage for GABA + ACh versus no drugs. **h** GABA (10 μM) effects on KCNQ2/3 in oocytes are not prevented by partial depletion/inhibition of synthesis of PIP$_2$ using wortmannin (30 μM for 3 h) ($n = 5$). Left, mean tail current; right, GABA dose response at −60 mV; ($n = 5$). **i** GABA (100 μM) has no effect on KCNQ1/KCNE1. Left, averaged traces; right, mean tail currents ($n = 6$). **j** GABA (100 μM) does not alter resting membrane potential ($E_M$) of unclamped oocytes expressing KCNQ1-KCNE1 ($n = 6$)

present in neuronal KCNQs but not cardiac-expressed KCNQ1, to be developed by pharmaceutical companies. There are other mechanisms for ligand-dependent opening of KCNQs, however, as some pharmacological agents activate KCNQ1, which lacks the S5 Trp, or activate neuronal KCNQs even after their S5 Trp has been substituted[38]. Conversely, it is interesting to note that retigabine itself is a weak activator of GABA receptors[39], perhaps a further hint of similarities between the W265-based binding site in KCNQ3 and the GABA-binding site of canonical GABA$_A$ receptors. It will be of interest to determine whether cnidarian KCNQs bearing the S5 W are also sensitive to GABA, and to compare their expression pattern to that of non-W cnidarian KCNQs.

Our principal finding, that M-channels are novel, unexpected targets for GABA, with sensitivity comparable to canonical GABA$_A$Rs in vertebrate CNS, changes current understanding of

the potential mechanisms underlying GABA-mediated inhibition, which up until now was considered to involve solely GABA$_{A, B, C}$ receptors. The EC$_{50}$ we observe for KCNQ2/3-dependent cellular hyperpolarization of 120 nM GABA (Fig. 2h), is similar both to the Kd we observe for $^3$H-GABA binding to KCNQ3 (Fig. 1j), and to the tonic level of GABA (160 nM), consistent with KCNQ2/3 channels directly binding GABA to influence neuronal membrane potentials at rest. Our findings, together with the known ability of KCNQ2/3 to regulate GABA release[8, 9], and strong expression overlap of KCNQ2/3 with GABAergic neurons[7], suggest the potential for an underlying feedback mechanism.

Additionally, our results demonstrate that KCNQ2/3 has the capacity to act as a chemosensor of the extracellular neurotransmitter/metabolite landscape, which could enable M-channels to respond to the balance of these molecules and by doing so regulate cellular excitability over time. Metabolic shifts favoring

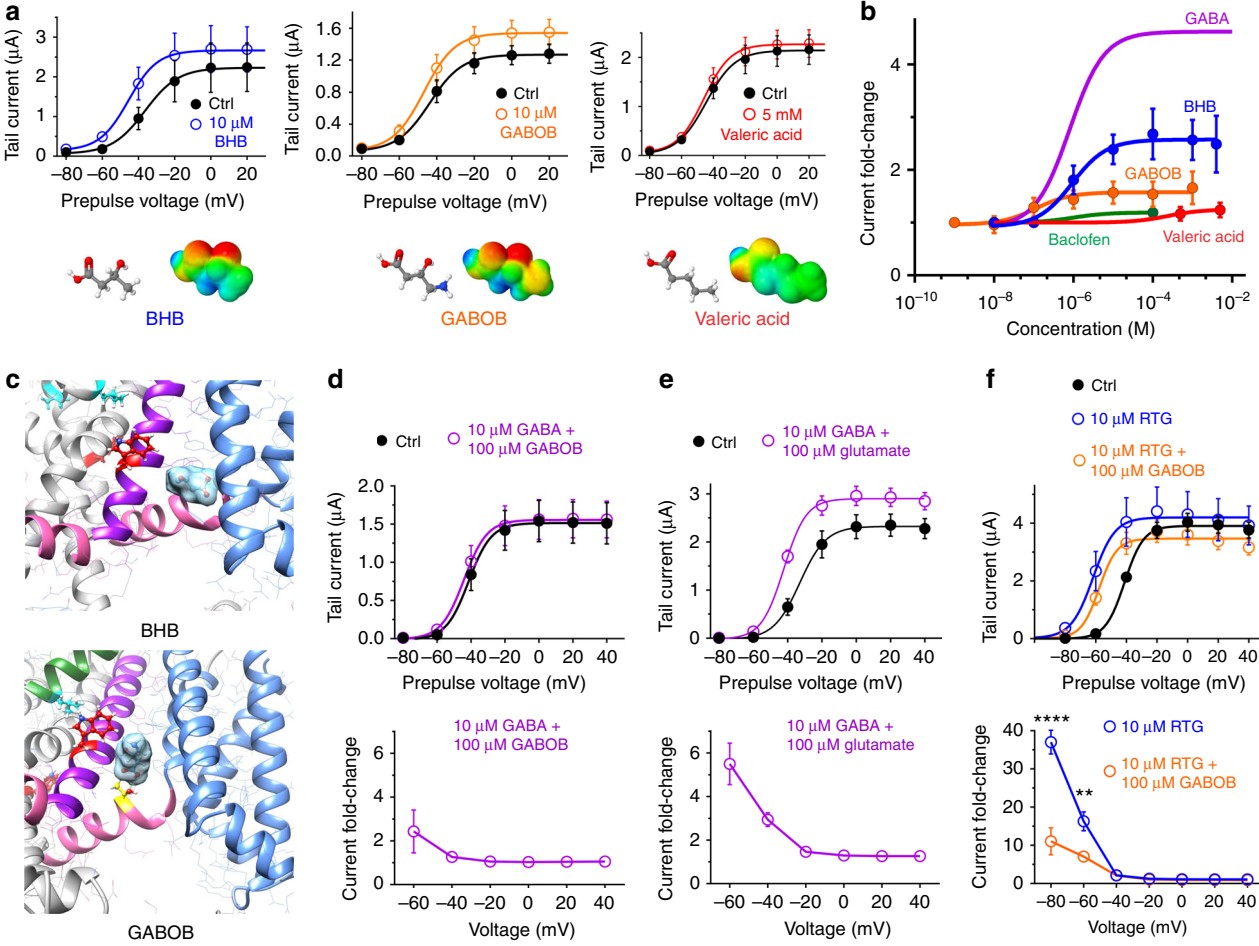

**Fig. 6** GABA and metabolites compete for binding to KCNQ3-W265. All error bars in figure indicate SEM. **a** Mean effects of BHB and GABOB (10 μM) versus valeric acid (no effects at 5 mM) on KCNQ2/3 in oocytes ($n = 5$-6). **b** Mean dose response of KCNQ2/3 activation at −60 mV by GABA (from Fig. 2) and related compounds ($n = 5$-10). **c** In silico docking predicts that BHB and GABOB bind to KCNQ3-W265, the latter also H-bonding with L237 in the S4–5 linker. **d**, **e** GABOB (**d**), but not glutamate (**e**), antagonizes the effects of GABA on KCNQ2/3 activation in oocytes ($n = 4$-6). Upper, mean tail currents; lower, current fold-change elicited by neurotransmitter combination versus voltage. **f** GABOB antagonizes the effects of retigabine on KCNQ2/3 activation in oocytes ($n = 10$). Upper, mean tail currents; lower, current fold-change elicited versus voltage. Data are for KCNQ2/3 in the absence of retigabine (Ctrl), with retigabine (RTG), or with retigabine + GABOB (RTG + GABOB). **$P < 0.01$; ****$P < 0.0001$

increased extracellular GABOB over GABA would be predicted, based on our data, to favor cellular excitability, as GABOB could compete out GABA from KCNQ3-W265, resulting in a positive shift in the voltage dependence of M-current activation compared to if GABA were bound (as in Fig. 6d). It is also interesting to speculate a role for BHB activation of KCNQ2/3 channels in the anticonvulsant effects of the ketogenic diet, given that BHB is the first ketone body produced during fasting or ketogenic diets, and also in diabetic ketoacidosis. However, all these ideas require further testing in native brain tissue preparations before any firm conclusions can be drawn as to the precise physiological roles of M-current GABA sensitivity. Finally, as GABA and analogs, in particular GABOB (EC$_{50}$ of 0.12 μM), activate KCNQ2/3 channels with a greater potency (albeit lower efficacy) than retigabine and analogs, the new findings could point the way to alternative chemical spaces for therapeutic neuronal KCNQ activation.

## Methods
**Channel subunit cRNA preparation and oocyte injection**. cRNA transcripts encoding human was KCNQ1, KCNQ2, KCNQ3, KCNQ4, KCNQ5, KCNA1, or

KCNE1 were generated by in vitro transcription using the T7 polymerase mMessage mMachine kit (Thermo Fisher Scientific), after vector linearization, from complementary DNA (cDNA) sub-cloned into plasmids incorporating *Xenopus laevis* β-globin 5′ and 3′ untranslated regions flanking the coding region to enhance translation and cRNA stability. cRNA was quantified by spectrophotometry. Mutant KCNQ3 cDNAs were generated by site-directed mutagenesis using a QuikChange kit according to manufacturer's protocol (Stratagene, San Diego, CA) and corresponding cRNAs prepared as above. Defolliculated stage V and VI *Xenopus laevis* oocytes (Ecocyte Bioscience, Austin, TX) were injected with Kv channel α subunit cRNAs (5–10 ng). Oocytes were incubated at 16 °C in Barth's saline solution (Ecocyte) containing penicillin and streptomycin, with daily washing, for 3–5 days prior to TEVC recording or $^3$H-GABA binding assays.

**Radioligand binding assays**. Each group of oocytes was placed in a round-bottomed, 10-ml culture tube, washed with ND96, and then resuspended in ND96 containing 100 nM γ-[2,3-$^3$H(N)]-aminobutyric acid ($^3$H-GABA) (Perkin Elmer, Waltham, MA) at 25–40 Ci/mmol specific activity, for a 30 min incubation at room temperature. Oocytes were then washed four times in 16 °C ND96, then transferred to individual wells in a 96-well plate and lysed in 0.2% SDS in ND96. Each oocyte lysate was transferred to a scintillation vial containing 5 ml cytoscint scintillation cocktail fluid (MP Biomedicals, Santa Ana, CA). Vials were capped, shaken, and then allowed to sit at room temperature for at least 30 min before scintillation counting in a Beckmann Coulter LS6500 liquid scintillation counter. Counts per minute were normalized to channel protein surface expression, which was

quantified using a surface biotinylation approach similar to previously described for oocyte membrane proteins[40] as follows: in parallel experiments to the binding assays, at 5 days post-cRNA injection, *Xenopus* oocytes (5 per group) expressing KCNQ1, 2, 3, 4 or 5, and water-injected controls, were washed with ND96, and gently agitated for 1 h at 4 °C in sulfo-NHS-SS-biotin (Pierce) (0.5 mg/ml in ND96). Oocytes were next washed in 5 ml per group quenching solution (192 mM glycine, 25 mM Tris-HCl, pH 7.4) for 10 min to quench unreacted biotin. Oocytes were next incubated in 1 ml per group homogenization buffer (in mM, 150 NaCl, 2 CaCl₂, 20 Tris, 2% Nonidet P-40) with 2% protease inhibitor cocktail (Pierce) and then lysed with a 20 ml pipette tip, vortexed, centrifuged at 4000 r.p.m. (10 min, 4 °C). The supernatant was mixed in Eppendorf tubes end-over-end overnight at 4 °C with high-capacity streptavidin agarose beads (Thermo Fisher, Huntington Beach, CA) (20 μl per group). The beads were gently centrifuged to precipitate surface-biotinylated membrane proteins that were bound to the avidin-coated beads, washed four times in homogenization buffer, resuspended in SDS-PAGE loading buffer containing 5% dithiothreitol, heated to 70 °C for 10 min with intermittent vortexing, and then separated on 4–12% ClearPage TEO-Tricine gradient gels (CBS Scientific, San Diego, CA). Gels were stained for 20 min in Coomassie Brilliant Blue G-250 (Sigma) in methanol/acetic acid, then destained for several hours in methanol/acetic acid. Coomassie blue was used, as its staining versus protein concentration is linear over a wide protein concentration range, membrane proteins are relatively scarce in non-injected oocyte membranes, and this approach avoids epitope tagging the different KCNQs, which could lead to other artifacts such as altered GABA binding or surface expression. There was one prominent protein band in the gel lanes of KCNQ-expressing groups, and a similar but much fainter band in non-injected oocytes (perhaps corresponding to endogenous *Xenopus* KCNQ1[41]). Band densities were quantified using ImageJ (NIH, Bethesda, MD), averaged for four groups of five oocytes each, and then binding assay ³H-GABA CPM were normalized to mean band density to correct for the different surface expression of the different KCNQ isoforms. This correction did not qualitatively alter the pattern of results, which in either corrected or non-corrected datasets showed a clear distinction between binding in water or KCNQ1-cRNA-injected oocytes versus the much higher binding in oocytes expressing KCNQ2–5 (Supplementary Fig. 1).

For the binding competition assay, the control group was treated as before. The competition group was placed in a round-bottomed, 10-ml culture tube, washed with ND96 containing 1 mM cold γ-aminobutyric acid (Sigma, Carlsbad, CA) and resuspended in ND96 containing 100 nM γ-[2,3-³H(N)]-aminobutyric acid and 1 mM cold GABA (Sigma), and incubated for 30 min at room temperature. The competition group was then washed four times in 16 °C ND96 containing 1 mM γ-aminobutyric acid, then transferred to individual wells in a 96-well plate and lysed in 0.2% SDS in ND96. Each oocyte lysate was transferred to a scintillation vial containing 5 ml cytoscint scintillation cocktail fluid (MP Biomedicals). Vials were capped, shaken, and then allowed to sit at room temperature for at least 30 min before scintillation counting in a Beckmann Coulter LS6500 liquid scintillation counter. For the saturation binding assays, a similar approach was taken but a range of ten different γ-[2,3-³H(N)]-aminobutyric acid concentrations were employed (0.1–1000 nM), with versus without 1 mM cold GABA. The groups with cold GABA in addition to γ-[2,3-³H(N)]-aminobutyric acid were used to calculate non-specific binding, and subtracted from total binding to give specific binding. CPM were converted to fmol/oocyte after correcting for counter efficiency. $B_{max}$ and $K_d$ values were calculated using automated nonlinear regression analysis in GraphPad prism.

**Two-electrode voltage clamp.** TEVC recording was performed at room temperature with an OC-725C amplifier (Warner Instruments, Hamden, CT) and pClamp8 software (Molecular Devices, Sunnyvale, CA) 3–5 days after cRNA injection as described in section above. Oocytes were placed in a small-volume oocyte bath (Warner) and viewed with a dissection microscope. Chemicals were sourced from Sigma. Bath solution was (in mM): 96 NaCl, 4 KCl, 1 MgCl₂, 1 CaCl₂, 10 HEPES (pH 7.6). GABA and GABOB were stored at −80 °C as 1 M stocks in molecular grade H₂O and diluted to working concentrations on each experimental day. Valeric acid was stored as a 5 mM stock in Ringer's solution at 4 °C and diluted to working concentrations each experimental day. β-hydroxybutyric acid was stored at 4 °C as a 480 mM stock in 100% ethanol and diluted to working concentrations each experimental day. All compounds were introduced to the recording bath via gravity perfusion at a constant flow of 1 ml per min for 3 min prior to recording. Pipettes were of 1–2 MΩ resistance when filled with 3 M KCl. Currents were recorded in response to pulses between −80 mV and +40 mV at 20 mV intervals, or a single pulse to +40 mV, from a holding potential of −80 mV, to yield current–voltage relationships, current magnitude, and for quantifying activation rate. Deactivation was recorded at −80 mV after a single activating pulse to +40 mV, from a holding potential of −80 mV. For wortmannin studies, oocytes were incubated in 30 μM wortmannin for 3 h at 16 °C prior to recording. TEVC data analysis was performed with Clampfit (Molecular Devices) and Graphpad Prism software (GraphPad, San Diego, CA, USA); values are stated as mean ± SEM. Normalized tail currents were plotted versus prepulse voltage and fitted with a single Boltzmann function:

$$g = \frac{(A_1 - A_2)}{\left\{ 1 + \exp\left[ \left( V_{\frac{1}{2}} - V \right) / V_S \right] \right\}} + A_2, \qquad (1)$$

where $g$ is the normalized tail conductance, $A_1$ is the initial value at $-\infty$, $A_2$ is the final value at $+\infty$, $V_{1/2}$ is the half-maximal voltage of activation, and $V_S$ the slope factor. Activation and deactivation kinetics were fitted with single exponential functions.

**Whole-cell patch clamp electrophysiology.** CHO cells (ATCC) were seeded on poly-L-lysine-treated glass coverslips in 24-well plates and transfected the next day using 0.5 μg each of mouse KCNQ2 and KCNQ3 cDNA in a CMV vector per well in normal media (DMEM with 10% FBS and 1% penicillin/streptomycin). The transfection reagent used was TransIT-LT1 (Mirus Bio LLC, Madison, WI, USA) and cells were patched 48–72 h post transfection. PC12 cells (Sigma) were grown in a 95% O₂/5% CO₂ and humidified environment at 37 °C in RPMI 1640 medium supplemented with 10% heat inactivated horse serum, 5% fetal bovine serum, and 1% penicillin/streptomycin. Media was replenished every 3 days and cultures were split once a week. For differentiation, cells were plated on 13 mm plastic coverslips, pre-treated with poly-L-lysine and collagen, in RPMI 1640 medium supplemented with 1% horse serum and 50 ng/ml nerve growth factor (NGF) and cultured for up to 7 days. Neurite outgrowth was observed from days 3–4 of differentiation and cells were used for experiments from 5 days post differentiation. Cell culture plastic ware and reagents were purchased from VWR and Fisher Scientific unless otherwise stated.

For both CHO cells and PC12 cells, currents were recorded in whole-cell mode at room temperature (22–25 °C) with 3–6 MΩ borosilicate glass electrodes backfilled with solution containing (in mM): 90 K acetate, 20 KCl, 40 HEPES, 3 MgCl₂, 1 CaCl₂, 3 EGTA-KOH, 2 MgATP; pH 7.2. Cells were continuously perfused at 1–2 ml/min with extracellular solution (ECS) containing (in mM): 135 NaCl, 5 KCl, 5 HEPES, 1.2 MgCl₂, 2.5 CaCl₂, 10 glucose; pH 7.4. Drugs were added to the ECS where indicated, as follows: 100 μM picrotoxin, 10 μM CGP55845, 10 μM XE991, and 100 μM GABA. All chemicals were purchased from Fisher Scientific or Sigma-Millipore.

Cells were held at −80 mV in voltage clamp before applying the voltage step protocols and currents were recorded, in response to pulses between −80 mV and +40 or +60 mV at 20 mV intervals followed by a single pulse to −30 mV, using a CV −7A Headstage (Axon Instruments, Foster City, CA, USA). Currents were amplified using a Multi-clamp 700B (Axon Instruments), low-pass filtered at 2–10 kHz using an eight-pole Bessel filter and digitization was achieved (sampling at 10–40 kHz) through a DigiData 1322A interface (Molecular Devices). The pClamp8 (Molecular Devices) software package was used for data acquisition and Clampfit was used for analysis, along with Graphpad Prism 7.0 (Graphpad). Normalized tail currents were plotted versus prepulse voltage and fitted with a single Boltzmann function.

**Neuronal isolation and recordings.** Adult male mice were purchased from Jackson Labs and then housed and used according to a protocol approved by the Institutional Animal Care and Use Committee of University of California, Irvine, and consistent with recommendations in the NIH Guide for the Care and Use of Laboratory Animals (8th edition, 2011). To analyze the effects of GABA on native neuronal M-current, DRG were isolated from adult male mice (2–4 months of age), and dissociated using collagenase and trypsin. DRG neurons were then cultured on poly-L-lysine and laminin-coated plastic coverslips in RPMI 1640 with 10% horse serum and 1% penicillin/streptomycin at 37 °C for 1–4 days before electrophysiological recordings were performed using perforated patch clamp. Amphotericin B (VWR, Radnor, PA) was dissolved in DMSO at 5 mg/ml (stock solution), then diluted to 150–200 μg/ml in intracellular solution and sonicated with a probe sonicator on ice for 5 s, kept at room temperature in the dark, and replaced every hour with a fresh solution from the stock. The working amphotericin solution was backfilled into borosilicate glass electrodes (3–5 MΩ) previously tip-filled to ~0.5 mm with intracellular solution lacking amphotericin B, immediately prior to sealing onto individual neurons. After whole-cell configuration was achieved and capacitance stabilized, whole-cell recordings were performed essentially as described in the prior paragraph, with the following differences: neurons were held at −20 mV to inactivate other channels before the test pulses; offline leak subtraction was performed.

**Sequence analysis and modeling.** Chemical structures and electrostatic surface potentials (range, −0.1 to 0.1) were plotted using Jmol, an open-source Java viewer for chemical structures in 3D: http://jmol.org/. For docking, the *X. laevis* KCNQ1 cryoEM structure[42] was first altered to incorporate KCNQ3/KCNQ5 residues known to be important for retigabine and ML-213 binding, and their immediate neighbors, followed by energy minimization using the GROMOS 43B1 force field[43], in DeepView[44]. Thus, *X. laevis* KCNQ1 amino acid sequence LIT*TL*YIGF was converted to LIT*AW*YIGF, the underlined W being W265 in human KCNQ3/KCNQ5 and the italicized residues being the immediate neighbors in KCNQ3/KCNQ5. In addition, *X. laevis* KCNQ1 sequence WWG*VVTVT*TIGYGD was converted to WWG*LITLA*TIGYGD, the underlined L being Leu314 in human KCNQ3/KCNQ5 and the italicized residues being the immediate neighbors in KCNQ5 and/or KCNQ3. Surrounding non-mutated sequences are shown to illustrate the otherwise high sequence identity in these stretches. Unguided docking of GABA, retigabine, and ML-213, to predict native binding sites, was performed

using SwissDock[45] with CHARMM forcefields[46]. Phylogenetic analysis was performed using BLAST searching[47] of predicted or known protein sequences from multiple genomes (where available) from each clade; KCNQ5 sequences for representative organisms from each clade are shown. Genomes were accessed using the US National Center for Biotechnology Information (NCBI), Joint Genome Institute (JGI), Kyoto Encyclopedia of Genes and Genomes (KEGG), UCSC Genome Browser Gateway, and the Elephant Shark Genome Project. Species names for the exemplars shown in Fig. 1a are as follows: Human, *Homo sapiens*; Frog, *Xenopus tropicalis*; Carp, *Cyprinus carpio*; Shark, *Callorhincus milii*; Lamprey, *Petromyzon marinus*; Acorn worm, *Sacclogossus kowalevskii*; Bat star, *Patiria miniata*; Snail, *Biomphalaria glabrata*; Fly, *Musca domestica*; Leech, *Helobdella robusta*; Tapeworm, *Hymenolepsis microstoma*; Nematode, *Caenorhabditis elegans*; Hydrozoa, *Hydra vulgaris*; Anthozoa, *Nematostella vectensis*.

**Statistical analysis**. All values are expressed as mean ± SEM. One-way analysis of variance (ANOVA) was applied for all tests; if multiple comparisons were performed, a post-hoc Tukey's HSD test was performed following ANOVA. All P-values were two-directional. Statistical significance was defined as $P < 0.05$.

**Data availability**. The datasets generated during the current study are available from the corresponding author on reasonable request.

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

## Acknowledgements

This study was supported by the US National Institutes of Health (GM115189 to G.W. A.). We are grateful to Lily Chen and Angele De Silva (University of California, Irvine) for generating mutant channel constructs, and to Drs. Naoto Hoshi and Andrei Yeremin (University of California, Irvine) for helpful advice regarding DRG neuron isolation and recording. We thank Bo Abbott and Spot for the animal silhouettes in Fig. 1.

## Author contributions

R.W.M. performed the oocyte experiments and analyses, prepared most of the figure panels, and edited the manuscript. M.P. performed CHO, PC12, and DRG neuron electrophysiology, analyzed data, and edited the manuscript and figures. G.W.A. conceived the study, performed biochemical experiments, sequence and structural analyses, wrote the manuscript, and prepared the figures.

## Additional information

**Competing interests:** The authors declare no competing interests.

