## [Peer Review File · Nature Communications]

Reviewers' comments:

Reviewer #1 (Remarks to the Author):

Thank you for the invitation to review the article 'Direct neurotransmitter activation of voltage-gated potassium channels' by Manville and colleagues. This is an interesting study that describes the activation of certain KCNQ channel types by GABA and other naturally occurring metabolites. The investigation is framed as a search for endogenous ligands of the channel that may target a pore site that has been shown to bind retigabine and many other KCNQ activators, but has no known physiological function. Using a radioligand binding assay and electrophysiology, the authors demonstrate that GABA binds to the neuronal KCNQ2,3,4,5 isoforms, but not KCNQ1, and has significant gating effects on the KCNQ3 and KCNQ5 isoforms. The authors go to extensive lengths to rule out other possible mechanisms of activation related to GABA effects on other signaling cascades, with appropriate pharmacological controls. They also demonstrate that GABA activation of KCNQ2/3 modulates its sensitivity to PIP2, an important consideration in terms of physiological regulation of these channels by GPCR signaling. Overall, I suspect this study would be of broad interest as it demonstrates an unexpected effect of GABA on voltage-gated potassium channels. This effect may be relevant for signaling in the nervous system and also KCNQ channel pharmacology (anti-epileptic drugs). I have a few comments that I hope will be constructive and helpful.

1. The evolutionary information is useful and appreciated in terms of emergence of the pore Trp that forms the retigabine binding site. However, the authors may wish to add that there appears to be a variety of mechanisms for KCNQ channel activators, as many retain their activity in channels after mutation of the pore Trp.
2. Line 255 'slow saturation... and resistance to washout... consistent with a longer-term regulation of neuronal excitability'. I would be cautious with this argument/statement in the absence of better experiments describing the interaction kinetics of the compounds with the channel, especially in *Xenopus* oocytes. Also, in vivo the time 'window' of GABA effects could depend strongly on local variation in mechanisms by which the transmitter is cleared.
3. I was very curious whether any of the transmitters could compete with retigabine. The efficacy of retigabine is much higher than described for GABA, BHB, and GABOB in certain subunit combinations, but the apparent affinity for some of the metabolites suggests that they could influence the retigabine effect. This potentially important pharmacological effect might be a very important outcome of the authors' main finding.
4. Methods line 346 – I believe the equation is not correct as written?
5. This may be tangential, but it may be worthwhile to point out that retigabine has also been shown to weakly activate GABA receptors.
6. I would encourage the authors to significantly pare down the supplemental information. It is a lot to get through and a lot of the data presented was quite redundant and did not add to the main arguments of the paper.

Reviewer #2 (Remarks to the Author):

In this manuscript, the Abbott's team used a phylogenetic analysis and in silico docking to probe the hypothesis that the anticonvulsant retigabine binding pocket containing the crucial tryptophan (W265 in human KCNQ3, W270 in KCNQ5) evolved as a chemosensor for the GABA native neurotransmitter ligand present in both primitive and modern nervous systems. Radioligand binding assays reveal that the anticonvulsant binding pocket of KCNQ2-Q5 can bind [H3] GABA. However, electrophysiological data indicate that the activity of KCNQ1, 2 and 4 was essentially GABA-insensitive, where GABA could increase only KCNQ3 and KCNQ5 channel amplitude. GABA and endogenous metabolites 27 β -hydroxybutyric acid (BHB) and γ -amino- β -hydroxybutyric acid (GABOB), competitively and differentially shift the voltage dependence of KCNQ3 activation. Based on their experimental data, the authors propose a novel mechanism, where the GABA neurotransmitter in addition to its effects on GABA receptors (ionotropic and metabotropic), can directly activate endogenous voltage-gated KCNQ3 and KCNQ5 channels to regulate neuronal excitability. This manuscript raises stimulating questions and reveals potentially interesting data, but one becomes very quickly disappointed by the lack of solid experimental data that could provide a basis for the conclusions drawn by the authors. The results depicted in this work suffer from important experimental weaknesses that severely prevent the publication of this manuscript. I have major concerns that are outlined below:

1-The radioligand binding assay (Fig. 1h) of [H3] GABA performed in injected *Xenopus* oocytes was carried out without checking the specific binding by measuring [H3] GABA in the absence and presence of excess cold GABA. Why? Neither saturation curves nor derived Scatchard plots were provided to yield apparent K_d affinity values and to examine whether GABA binds to a single binding site in KCNQ channels. These data should be provided to convince the reviewer about the specificity of GABA interaction with the anticonvulsant binding pocket. Also, in the discussion, the authors do not provide any rational explanation of how GABA does not activate KCNQ4 currents, while KCNQ4 has an identical sequence in the anticonvulsant binding pocket to that of KCNQ3 and KCNQ5.

2- In Figure 3g,h, the normalized conductance-voltage relations were calculated from the tail currents. However, the tail currents were measured at a repolarizing potential of -30 mV, which is very problematic. Usually, to derive such G/V relations, one needs to repolarize the tail potential to negative values that reflect the transition of the open state to the closed state (O to C transition). The values of the tail potential provided in this manuscript at -30 mV do not reflect such a transition, since at -30 mV a substantial population of the KCNQ channels is open. Consequently, the G/V relations and the resulting V_{50} values are usually biased towards more negative potentials. Related to this experimental problem, one cannot measure appropriately deactivating currents and their related time constants. One cannot appreciate how GABA slows down the deactivating currents.

3-To appreciate the impact of GABA on native M-currents, the authors chose a neuronal cells line, PC12 cells (Figure 4c,d). The data provided by the authors are not convincing and it is difficult to appreciate the contribution of M-currents in this cell line that express high densities of other contaminating K^+ channels such as Kv2.1 and others. In the labs of Hille, Shapiro, Brown and many others, they rather used the sympathetic ganglion neurons or the small dorsal root ganglion neurons to measure the M-currents using the deactivation tail protocol (holding to -20 mV and repolarizing to -60 mV) in the absence and presence of the M-channel blocker XE-991. The authors should use these preparations and protocol to

convince this reviewer.

4-In CHO cells, the V50 values for KCNQ2/Q3 without GABA are quite depolarized (-2.9 mV). Why? Usually, the V50 ranges between -20 mV to -30 mV.

5- Most importantly, there is no attempt by the authors to provide data from a neurophysiological approach showing that presynaptic activation of a GABAergic neuron (for example in hippocampal slice) leads to enhanced M-current amplitude measured in a postsynaptic pyramidal neuron (e.g., CA1) in the presence of GABAA and GABAB receptor blockers. Instead, the authors provide uninteresting data on Xenopus expression system to prove that the effects of GABA do not involve endogenous channels or endogenous muscarinic receptors. This issue could be quickly solved if the authors would choose exclusively the CHO expression system. To support their very speculative claim that GABA in addition to its effects on GABA receptors, can directly activate voltage-gated KCNQ3 and KCNQ5 channels to regulate neuronal excitability, the authors should provide neurophysiological data, as exemplified above. In UC Davis, there are many neurophysiologists who could provide such collaborative experimental help.

In conclusion, the authors propose a novel mechanism, where GABA in addition to its effects on GABA receptors, can directly activate voltage-gated KCNQ3 and KCNQ5 channels to regulate neuronal excitability. Unfortunately, the experimental data provided in this manuscript do not support this claim. Instead of providing supportive neurophysiological evidence, the authors make heroic efforts essentially focused on the Xenopus oocyte expression system, which has little relevance for the main question addressed in the manuscript.

Reviewer #3 (Remarks to the Author):

The authors demonstrated in this study that GABA directly binds to the Tryptophan residue in TM5 of KCNQ2,3,4,5 channels and increases the activity of KCNQ3 and 5 channels at low concentration, ruling out all other possibilities by strong evidences based on data by ligand binding assay, in silico molecular docking and electrophysiological analyses. The effect was observed not only in heterologous expression system such as Xenopus oocytes and CHO cells, but also confirmed in differentiated PC12 cells. Also, the authors showed the effect of GABA overcomes the inhibitory effect of PIP2 breakdown, and examined the effect of GABA metabolites, such as GABOB. The evolutionary aspect was carefully analyzed and discussed, demonstrating the significance of the present finding and elevating the charm of this paper.

Overall, this reviewer was very much impressed by the truly high impact of the findings, thorough performance of various intensive experiments and high quality data which convincingly consolidate the conclusion. The manuscript is written carefully and it is polished up to be concise and clear-cut. I did not find any major flaws and believe the manuscript can be accepted even in the present form. In conclusion, I judge this manuscript should be published with a very high priority in nature communications.

I have some minor comments just for reference.

1. In Fig. 4b, d (patch clamp data), tail currents are normalized and the shift of $V_{1/2}$ is described in the main text, but not in Figs. 2, 3, 5 (TEV data). Thus, there is an impression of inconsistency in the way of the analyses and presentations. It would be better to show non-normalized tail current plot in Fig. 4b, d, and also to describe the shift of $V_{1/2}$ in the main text about KCNQ3 data (Fig. 2a) and KCNQ2/3 data (Fig. 2d), by adding normalized tail current data in Fig. 2a and 2d, or by describing the data in e.g. Suppl Figs 2-3 and 2-6.

2. Suppl. Tables: I cannot understand well why the $V_{1/2}$ values of the normalized and non-normalized tail currents are different. In principle, they should be identical.

3. Have the authors tried L to W mutation of KCNQ1?

4. The evolutionary timing when KCNQ channel acquired sensitivity to GABA is carefully discussed. To appreciate the discussion better, I am curious to know when GABA (and GAD, its synthetase) and GABA receptors appeared in the phylogenic tree.

5. Although the effect of GABA on KCNQ channels was studied in differentiated PC12 cells, it is useful to confirm that the effect can be observed also in neurons in primary culture and/or in brain slice preparation, in the presence of blockers of GABAA receptor and GABAB receptor. Furthermore, it is obviously beneficial to analyze the behavioral abnormalities of KCNQ3 W265L knock-in mice to prove the physiological relevance of the findings. However, I think these data are NOT essential for publication of the present paper, and rather think they are important must-do tasks at the next stage. I judge the present study deserves immediate publication, without adding data of these time consuming experiments.

Response to Reviewers

Reviewers' comments:

Reviewer #1 (Remarks to the Author):

Thank you for the invitation to review the article 'Direct neurotransmitter activation of voltage-gated potassium channels' by Manville and colleagues. This is an interesting study that describes the activation of certain KCNQ channel types by GABA and other naturally occurring metabolites. The investigation is framed as a search for endogenous ligands of the channel that may target a pore site that has been shown to bind retigabine and many other KCNQ activators, but has no known physiological function. Using a radioligand binding assay and electrophysiology, the authors demonstrate that GABA binds to the neuronal KCNQ2,3,4,5 isoforms, but not KCNQ1, and has significant gating effects on the KCNQ3 and KCNQ5 isoforms. The authors go to extensive lengths to rule out other possible mechanisms of activation related to GABA effects on other signaling cascades, with appropriate pharmacological controls. They also demonstrate that GABA activation of KCNQ2/3 modulates its sensitivity to PIP2, an important consideration in terms of physiological regulation of these channels by GPCR signaling. Overall, I suspect this study would be of broad interest as it demonstrates an unexpected effect of GABA on voltage-gated potassium channels. This effect may be relevant for signaling in the nervous system and also KCNQ channel pharmacology (anti-epileptic drugs). I have a few comments that I hope will be constructive and helpful.

1. The evolutionary information is useful and appreciated in terms of emergence of the pore Trp that forms the retigabine binding site. However, the authors may wish to add that there appears to be a variety of mechanisms for KCNQ channel activators, as many retain their activity in channels after mutation of the pore Trp.

- *We have added the following statement, as requested, to page 11: "There are other mechanisms for ligand-dependent opening of KCNQs, however, as some pharmacological agents activate KCNQ1, which lacks the S5 Trp, or activate neuronal KCNQs even after their S5 Trp has been substituted"*

2. Line 255 'slow saturation... and resistance to washout... consistent with a longer-term regulation of neuronal excitability'. I would be cautious with this argument/statement in the absence of better experiments describing the interaction kinetics of the compounds with the channel, especially in *Xenopus* oocytes. Also, in vivo the time 'window' of GABA effects could depend strongly on local variation in mechanisms by which the transmitter is cleared.

- *We have removed the statement as requested.*

3. I was very curious whether any of the transmitters could compete with retigabine. The efficacy of retigabine is much higher than described for GABA, BHB, and GABOB in certain subunit combinations, but the apparent affinity for some of the metabolites suggests

that they could influence the retigabine effect. This potentially important pharmacological effect might be a very important outcome of the authors' main finding.

- *We have now performed this experiment, and as predicted by the reviewer, we found that GABOB does indeed partially compete with retigabine, lowering its efficacy at -60 mV almost fourfold (Fig. 6 f).*

-

4. Methods line 346 – I believe the equation is not correct as written?

- *We have changed the equation for the version more commonly used.*

5. This may be tangential, but it may be worthwhile to point out that retigabine has also been shown to weakly activate GABA receptors.

- *We now include this statement: “Conversely, it is interesting to note that retigabine itself is a weak activator of GABA receptors³⁹, perhaps a further hint of similarities between the W265-based binding site in KCNQ3 and the GABA binding site of canonical GABA_A receptors.” (page 12, paragraph 1).*

6. I would encourage the authors to significantly pare down the supplemental information. It is a lot to get through and a lot of the data presented was quite redundant and did not add to the main arguments of the paper.

- *We tried to be inclusive with the data in the interests of rigor and reproducibility. Perhaps the editor can indicate the journal's preference of whether the data is best left in Supplementary Data or uploaded to a separate server.*

Reviewer #2 (Remarks to the Author):

In this manuscript, the Abbott's team used a phylogenetic analysis and in silico docking to probe the hypothesis that the anticonvulsant retigabine binding pocket containing the crucial tryptophan (W265 in human KCNQ3, W270 in KCNQ5) evolved as a chemosensor for the GABA native neurotransmitter ligand present in both primitive and modern nervous systems. Radioligand binding assays reveal that the anticonvulsant binding pocket of KCNQ2-Q5 can bind [³H] GABA. However, electrophysiological data indicate that the activity of KCNQ1, 2 and 4 was essentially GABA-insensitive, where GABA could increase only KCNQ3 and KCNQ5 channel amplitude. GABA and endogenous metabolites 27 β-hydroxybutyric acid (BHB) and γ-amino-β-hydroxybutyric acid (GABOB), competitively and differentially shift the voltage dependence of KCNQ3 activation. Based on their experimental data, the authors propose a novel mechanism, where the GABA neurotransmitter in addition to its effects on GABA receptors

(ionotropic and metabotropic), can directly activate endogenous voltage-gated KCNQ3 and KCNQ5 channels to regulate neuronal excitability.

This manuscript raises stimulating questions and reveals potentially interesting data, but one becomes very quickly disappointed by the lack of solid experimental data that could provide a basis for the conclusions drawn by the authors. The results depicted in this work suffer

from important experimental weaknesses that severely prevent the publication of this manuscript. I have major concerns that are outlined below:

1-The radioligand binding assay (Fig. 1h) of [H3] GABA performed in injected *Xenopus* oocytes was carried out without checking the specific binding by measuring [H3] GABA in the absence and presence of excess cold GABA. Why? Neither saturation curves nor derived Scatchard plots were provided to yield apparent K_d affinity values and to examine whether GABA binds to a single binding site in KCNQ channels. These data should be provided to convince the reviewer about the specificity of GABA interaction with the anticonvulsant binding pocket. Also, in the discussion, the authors do not provide any rational explanation of how GABA does not activate KCNQ4 currents, while KCNQ4 has an identical sequence in the anticonvulsant binding pocket to that of KCNQ3 and KCNQ5.

- *We have now conducted competition studies using cold GABA as requested, and found that 3H-GABA is almost completely competed out by an excess of cold GABA (Fig. 1i).*
- *In addition, we performed saturation binding experiments, using ten different 3H-GABA concentrations, and parallel experiments using a similar set of GABA concentrations but with excess cold GABA to determine the nonspecific binding. This allowed us to calculate both a K_d and a B_{max} for GABA binding to KCNQ3 (Fig. 1j). These findings should satisfy any concerns regarding specific versus nonspecific binding.*
- *Re: explaining why GABA does not activate KCNQ4, the short stretch near the Trp is identical in these channels, but surrounding residues in the 3D channel structure are not. This is why retigabine shows different efficacy/affinity for the various KCNQ2-5 isoforms even though the W is essential for retigabine binding, as we find for GABA. We explain this in the Discussion (p11, paragraph 2).*

2- In Figure 3g,h, the normalized conductance-voltage relations were calculated from the tail currents. However, the tail currents were measured at a repolarizing potential of -30 mV, which is very problematic. Usually, to derive such G/V relations, one needs to repolarize the tail potential to negative values that reflect the transition of the open state to the closed state (O to C transition). The values of the tail potential provided in this manuscript at -30 mV do not reflect such a transition, since at -30 mV a substantial population of the KCNQ channels is open. Consequently, the G/V relations and the resulting V_{50} values are usually biased towards more negative potentials. Related to this experimental problem, one cannot measure appropriately deactivating currents and their related time constants. One cannot appreciate how GABA slows down the deactivating currents.

- *The assertion that voltage dependence cannot be accurately quantified using a -30 mV tail current is incorrect. As shown in, e.g., Figure 2c, we measure current at the arrow, a time point at which the channels have not yet had a chance to open in response to the tail current switch, because at -30 mV they open very slowly. Ample evidence for this is that the channels are almost all closed at -30 mV following a -80 mV prepulse (e.g., see Figure 2e). If one chooses a much more negative tail pulse, there are two problems: first, the channels close so quickly that the measurement reflects that closure, instead of how many were opened in the prepulse, which is the motivation behind using a tail current (to provide a snapshot of how many channels were opened at each voltage in the prepulse, but eliminating driving force effects on current magnitude). Second, because more negative voltages are closer to*

the K^+ equilibrium potential, the currents are much smaller, again reducing accuracy. Reviewer 2 noticed a negative-shifted voltage dependence specifically in Fig. 3 g,h, but that is due to the mutation(s), not the protocol. In fact, all the experiments were conducted similarly. Hence, use of the -30 mV tail current is standard practice in the KCNQ field.

- *We agree that it is preferable to quantify deactivation at more negative voltages. Thus, we repeated these studies using a +40 mV activation pulse followed by a -80 mV pulse during which we quantified deactivation rate. The new data are included in Fig. 2k.*

3-To appreciate the impact of GABA on native M-currents, the authors chose a neuronal cells line, PC12 cells (Figure 4c,d). The data provided by the authors are not convincing and it is difficult to appreciate the contribution of M-currents in this cell line that express high densities of other contaminating K^+ channels such as Kv2.1 and others. In the labs of Hille, Shapiro, Brown and many others, they rather used the sympathetic ganglion neurons or the small dorsal root ganglion neurons to measure the M-currents using the deactivation tail protocol (holding to -20 mV and repolarizing to -60 mV) in the absence and presence of the M-channel blocker XE-991. The authors should use these preparations and protocol to convince this reviewer.

- *To further support our findings we have conducted new studies using DRG neurons isolated from adult mice. Using perforated patch, we quantified the -60 mV current following a -20 mV holding potential in the presence of canonical GABA receptor blockers to remove the contribution of confounding currents. We observe an increase in the -60 mV current relative to the -20 mV current, as one would predict from our other data in the paper. We include this new data in Fig. 4e-h.*

4-In CHO cells, the V_{50} values for KCNQ2/Q3 without GABA are quite depolarized (-2.9 mV). Why? Usually, the V_{50} ranges between -20 mV to -30 mV.

- *Many factors, including PIP2 and specific ratio of KCNQ2 to KCNQ3, can influence $V_{0.5}$ values of KCNQ2/3 activation. We performed paired (+/- GABA on each cell) to ensure that, irrespective of variability in actual $V_{0.5}$, we quantified the relative effects of GABA on channel opening. In this respect, our CHO cell data matched well that of the PC12 cells and oocytes.*

5- Most importantly, there is no attempt by the authors to provide data from a neurophysiological approach showing that presynaptic activation of a GABAergic neuron (for example in hippocampal slice) leads to enhanced M-current amplitude measured in a postsynaptic pyramidal neuron (e.g., CA1) in the presence of GABAA and GABAB receptor blockers. Instead, the authors provide uninteresting data on Xenopus expression system to prove that the effects of GABA do not involve endogenous channels or endogenous muscarinic receptors. This issue could be quickly solved if the authors would choose exclusively the CHO expression system. To support their very speculative claim that GABA in addition to its effects on GABA receptors, can directly activate voltage-gated KCNQ3 and KCNQ5 channels to regulate neuronal excitability, the authors should provide neurophysiological data, as exemplified above. In UC Davis, there are many neurophysiologists who could provide such collaborative experimental

help.

In conclusion, the authors propose a novel mechanism, where GABA in addition to its effects on GABA receptors, can directly activate voltage-gated KCNQ3 and KCNQ5 channels to regulate neuronal excitability. Unfortunately, the experimental data provided in this manuscript do not support this claim. Instead of providing supportive neurophysiological evidence, the authors make heroic efforts essentially focused on the *Xenopus* oocyte expression system, which has little relevance for the main question addressed in the manuscript.

- *We used the reductionist approach, and an extremely stable system (*Xenopus* oocytes) for the majority of our studies because the main goals were to establish the identity of, and molecular basis for, a putative endogenous ligand for KCNQ2/3 channels. We achieved this, and much more, using *Xenopus* oocytes. These kinds of studies simply cannot be done in neurons, because of the confounding currents and many, many other issues. We have now further substantiated our findings using DRG neurons as requested, and also CHO and PC12 cells. We do not believe that this is a new mechanism for rapid synaptic neurotransmission. Rather, we believe that KCNQ2/3 channels may sense the tonic neurotransmitter/metabolite milieu to regulate neuronal excitability. Elucidating the precise physiological role for the fundamental finding that we have discovered and substantiated with many controls using oocytes, is a long-term project outside the scope of this paper (as alluded to by reviewers 1 and 3). There may not be a simple answer to this, and the field still does not know the precise role even for activation of canonical GABA receptors by tonic GABA levels outside the synapse. We consider it inappropriate to ask us to go above and beyond what is already known about the canonical receptors, in this first manuscript. We have started a new sub-field, identified the *K_d*, binding site, sensitivity, subunit specificity, efficacy, and even discovered a partial agonist. We believe that this, in addition to the DRG neuron recordings we have now performed, is enough for the first report. We understand that our work is a very new idea, and traditional ideas about canonical GABA receptors likely do not apply to this new system. However, in addition to the additional experiments, we have now further emphasized in the Discussion that the precise physiological roles require further elucidation, with the caveat in the final paragraph: “However, all these ideas require further testing in native brain tissue preparations before any firm conclusions can be drawn as to the precise physiological roles of M-current GABA sensitivity”*

Reviewer #3 (Remarks to the Author):

The authors demonstrated in this study that GABA directly binds to the Tryptophan residue in TM5 of KCNQ2,3,4,5 channels and increases the activity of KCNQ3 and 5 channels at low concentration, ruling out all other possibilities by strong evidences based on data by ligand binding assay, in silico molecular docking and electrophysiological analyses. The effect was observed not only in heterologous expression system such as *Xenopus* oocytes and CHO cells, but also confirmed in differentiated PC12 cells. Also, the authors showed the effect of GABA overcomes the inhibitory effect of PIP2 breakdown, and examined the effect of GABA metabolites, such as GABOB. The evolutionary aspect was carefully analyzed and discussed, demonstrating the significance of the present finding and elevating the charm of this paper.

Overall, this reviewer was very much impressed by the truly high impact of the findings, thorough performance of various intensive experiments and high quality data which convincingly consolidate the conclusion. The manuscript is written carefully and it is polished up to be concise and clear-cut. I did not find any major flaws and believe the manuscript can be accepted even in the present form. In conclusion, I judge this manuscript should be published with a very high priority in nature communications.

I have some minor comments just for reference.

1. In Fig. 4b, d (patch clamp data), tail currents are normalized and the shift of $V_{1/2}$ is described in the main text, but not in Figs. 2, 3, 5 (TEV data). Thus, there is an impression of inconsistency in the way of the analyses and presentations. It would be better to show non-normalized tail current plot in Fig. 4b, d, and also to describe the shift of $V_{1/2}$ in the main text about KCNQ3 data (Fig. 2a) and KCNQ2/3 data (Fig. 2d), by adding normalized tail current data in Fig. 2a and 2d, or by describing the data in e.g. Suppl Figs 2-3 and 2-6.

- *We have added a normalized KCNQ2/3 G/Gmax plot to fig 2c. We also now describe the normalized data for homomeric KCNQ3 and KCNQ5, and heteromeric KCNQ2/3, in the Results section (pages 4-5).*

2. Suppl. Tables: I cannot understand well why the $V_{1/2}$ values of the normalized and non-normalized tail currents are different. In principle, they should be identical.

- *The small differences relate to the fact that we fit the mean data to generate the $V_{1/2}$ values, because this produces a more accurate fit. Therefore, if a particular cell has greater raw tail current, its $V_{1/2}$ will have slightly more weight in the mean values than a cell with lower current. Hence, we report values for both the normalized and the non-normalized data. In addition, we noticed that some of the fits were not optimal and so we re-fitted and the data now more closely align. The standard errors are smaller for the normalized data and so those are the numbers we quote in the text.*

3. Have the authors tried L to W mutation of KCNQ1?

- *We have not tried that yet but it is on the list of things to do.*

4. The evolutionary timing when KCNQ channel acquired sensitivity to GABA is carefully discussed. To appreciate the discussion better, I am curious to know when GABA (and GAD, its synthetase) and GABA receptors appeared in the phylogenic tree.

We have added some text to discuss this in page 11 paragraph 1: "For comparison, Hydra are also known to express functional canonical GABA_A receptors with pharmacology similar in some respects to mammalian GABA_A receptors³⁴. GABA itself is found in all organisms; glutamate decarboxylase (GAD), which catalyzes conversion of glutamate into GABA, is found in bacteria (where decarboxylation of amino acids including glutamate aids

in acid resistance³⁵) and plants (e.g., in soybean³⁶) in addition to animals.” We also redid the phylogenetic tree on Figure 1 to better reflect current opinions about evolutionary relationships, generated our own animal silhouettes to avoid copyright issues, and uncovered an additional clade bearing the S5 Trp – members of the Cnidaria, including Hydra (discussed in the last paragraph of the Introduction).

5. Although the effect of GABA on KCNQ channels was studied in differentiated PC12 cells, it is useful to confirm that the effect can be observed also in neurons in primary culture and/or in brain slice preparation, in the presence of blockers of GABAA receptor and GABAB receptor. Furthermore, it is obviously beneficial to analyze the behavioral abnormalities of KCNQ3 W265L knock-in mice to prove the physiological relevance of the findings. However, I think these data are NOT essential for publication of the present paper, and rather think they are important must-do tasks at the next stage. I judge the present study deserves immediate publication, without adding data of these time consuming experiments.

- *We have added DRG neuron recordings to Fig. 4. Thanks for the comments - we agree that a logical next step is to generate the W265L mouse line using CRISPR-Cas9, and this is underway.*

Reviewers' comments:

Reviewer #1 (Remarks to the Author):

Thank you for the opportunity to have another look at this manuscript.

The authors have addressed all of my comments.

It would be good to clarify if the radioligand binding was done with KCNQ3[A315T] or KCNQ3 (eg. Figure 1 h,i,j), and perhaps acknowledge that the dissociation constant measured in the binding assay is stronger than suggested by electrophysiology.

Reviewer #2 (Remarks to the Author):

In their revised manuscript, the Abbott's team made laudable efforts to address the concerns of the reviewers in general and my concerns in particular. Unfortunately, the revised manuscript does not adequately address my concerns and still keeps me disappointed by the lack of experimental data that may provide a basis for the conclusions drawn by the authors. The results depicted in this revised version suffer from important weaknesses that prevent the publication of this manuscript. My major concerns are outlined below:

1- In the rebuttal "In addition, we performed saturation binding experiments, using ten different 3H-GABA concentrations, and parallel experiments using a similar set of GABA concentrations but with excess cold GABA to determine the nonspecific binding. This allowed us to calculate both a K_d and a B_{max} for GABA binding to KCNQ3 (Fig. 1j)".

In Figure 1j, the authors performed saturation binding experiments, using ten different 3H-GABA concentrations. In fact, one sees that only two concentrations of 3H-GABA are in the range of measurable binding. How were the authors able to calculate both a K_d and a B_{max} for 3HGABA binding using only two concentrations of measurable 3H-GABA binding? Which kind of Scatchard plot did they use to get a K_d and B_{max} with only two points?

2-In the rebuttal "The assertion that voltage dependence cannot be accurately quantified using a -30 mV tail current is incorrect".

I totally disagree with the authors on this issue. In their rebuttal, "Ample evidence for this is that the channels are almost all closed at -30 mV following a -80 mV prepulse". In Figure 2d (the right panel), one can clearly see in the conductance-voltage relations that at -30 mV, most of the M-channels are open and are even nearby the maximum open probability. There is enough electrochemical gradient to measure the tail currents at -60 mV (like many other labs, see David Brown and others), which should reflect the open to closed transition.

3-In the rebuttal "To further support our findings we have conducted new studies using DRG neurons isolated from adult mice. Using perforated patch, we quantified the -60 mV current following a -20 mV holding potential in the presence of canonical GABA receptor blockers to remove the contribution of confounding currents. We observe an increase in the -60 mV current relative to the -20 mV current, as one would predict from our other data in the paper. We include this new data in Fig. 4e-h".

This experiment is not convincing. To examine the physiological consequence of the possible interaction of GABA with the M-channels on neuronal excitability (DRG or central neurons) is

to record in the current-clamp configuration and in the presence of GABA_A and GABA_B blockers, the spike discharge with and without GABA.

In conclusion, the authors made a potentially interesting observation that M-channels could interact with GABA and may act as a sensor of the GABAergic milieu to regulate neuronal excitability. Unfortunately, the experimental data provided in this revised manuscript yet do not support this claim.

Reviewer #3 (Remarks to the Author):

The authors satisfactorily revised the manuscript in response to my previous comments, and I have no more specific comments. I judge this ms is ready for publication.

Response to Reviewers

Reviewers' comments:

Reviewer #1 (Remarks to the Author):

Thank you for the opportunity to have another look at this manuscript.

The authors have addressed all of my comments.

It would be good to clarify if the radioligand binding was done with KCNQ3[A315T] or KCNQ3 (eg. Figure 1 h,i,j), and perhaps acknowledge that the dissociation constant measured in the binding assay is stronger than suggested by electrophysiology.

- Thank you. The binding studies were performed using wild-type human KCNQ3. We now make this clearer in the manuscript (page 4, paragraph 1).

- We have added a sentence noting the difference between the binding Kd and the EC50 quantified when in voltage clamp (page 4, paragraph 2). Interestingly, the Kd in the binding assay was very close (126 nM versus 120 nM, respectively) to the EC₅₀ we observed for GABA effects on KCNQ2/3-dependent membrane potential (Fig. 2h), suggesting that when no voltage clamp is applied, the binding and gating potencies are similar. We mention this in the Results (page 4, paragraph 2) and Discussion (page 12, paragraph 2).

Reviewer #2 (Remarks to the Author):

In their revised manuscript, the Abbott's team made laudable efforts to address the concerns of the reviewers in general and my concerns in particular. Unfortunately, the revised manuscript does not adequately address my concerns and still keeps me disappointed by the lack of experimental data that may provide a basis for the conclusions drawn by the authors. The results depicted in this revised version suffer from important weaknesses that prevent the publication of this manuscript. My major concerns are outlined below:

1- In the rebuttal "In addition, we performed saturation binding experiments, using ten different 3H-GABA concentrations, and parallel experiments using a similar set of GABA concentrations but with

excess cold GABA to determine the nonspecific binding. This allowed us to calculate both a K_d and a B_{max} for GABA binding to KCNQ3 (Fig. 1j)".

In Figure 1j, the authors performed saturation binding experiments, using ten different 3H-GABA concentrations. In fact, one sees that only two concentrations of 3H-GABA are in the range of measurable binding. How were the authors able to calculate both a K_d and a B_{max} for 3HGABA binding using only two concentrations of measurable 3H-GABA binding? Which kind of Scatchard plot did they use to get a K_d and B_{max} with only two points?

- **As we previously stated, there are ten concentrations in the range of measurable binding. We now include an additional figure (new supplementary Figure 1 panel d) showing an expanded version of the low [GABA] portion of Fig. 1j, to more clearly show the points for binding at lower [GABA] values. Thus, we used 10 values to fit the curve and obtain K_d and B_{max} values. We did not use a Scatchard plot. Scatchard plots (actually more accurately attributed to Rosenthal) are little used now that nonlinear regression programs are readily available. This is because Scatchard plots are inaccurate, as the linear transformation distorts experimental data. According to Graphpad application notes, "linear regression assumes that the scatter of points around the line follows a Gaussian distribution and that the standard deviation is the same at every value of X. These assumptions are not true with the transformed data. A second problem is that the Scatchard transformation alters the relationship between X and Y. The value of X (bound) is used to calculate Y (bound/free) and this violates the assumptions of linear regression". Nonlinear regression is now considered to be a superior technique and, therefore, that is what we used to fit our specific binding curve, following subtraction of nonspecific binding using the cold GABA-competed values, as described in Materials and Methods.**

2-In the rebuttal" The assertion that voltage dependence cannot be accurately quantified using a -30 mV tail current is incorrect".

I totally disagree with the authors on this issue. In their rebuttal, "Ample evidence for this is that the channels are almost all closed at -30 mV following a -80 mV prepulse". In Figure 2d (the right panel), one can clearly see in the conductance-voltage relations that at -30 mV, most of the M-channels are open and are even nearby the maximum open probability. There is enough electrochemical gradient to measure the tail currents at -60 mV (like many other labs, see David Brown and others), which should reflect the open to closed transition.

- **Respectfully, I am afraid that Reviewer 2 does not understand tail protocols. In Figure 2e (incidentally, the reviewer incorrectly refers to the right panel of Figure 2d, which is not a tail protocol), the voltages refer to PREPULSE VOLTAGES. All the current values in Figure 2e (and all the tail protocols in this paper) were measured at -30 mV (at the beginning of the tail pulse, shown by the arrow in the voltage protocol diagram in the right hand side of Fig. 2e). As one can clearly see, at -30 mV, after a PREPULSE VOLTAGE step at -80 mV, there is essentially zero current in the control, because almost all the channels are closed. By measuring current at the arrow shown in the voltage protocol, the channels have not had time to be opened further by the -30 mV step, and so they are still closed. See areas highlighted in red in the figure below, which I have adapted from Fig 2 panels c and e.**

THIS IS STANDARD PRACTICE. I have published tens of papers using this protocol for KCNQ channels, as have many, many other labs. I can only imagine that Reviewer 2 is looking at the current level corresponding to the -30 mV *PREPULSE VOLTAGE* in Fig 2e, and confusing it with the -30 mV *TAIL PULSE VOLTAGE*. But the x axis is clearly labeled “Prepulse voltage”. This is also obvious in the current traces in Fig. 2c: at the arrow, the current is close to zero in the lowest trace, because the cells were prepulsed to -80 mV and then switched to a tail voltage of -30 mV, but the current was measured at the arrow, long before more of the channels were opened by the -30 mV step. Thus the current reflects how many channels were opened in the prepulse (-80 mV for the most negative prepulse) and not how many would eventually be opened at -30 mV if measured later on in the tail pulse. I am really sorry – I do not know how to make this explanation any simpler. I have never had to explain this to a reviewer before.

3-In the rebuttal “To further support our findings we have conducted new studies using DRG neurons isolated from adult mice. Using perforated patch, we quantified the -60 mV current following a -20 mV holding potential in the presence of canonical GABA receptor blockers to remove the contribution of confounding currents. We observe an increase in the -60 mV current relative to the -20 mV current, as one would predict from our other data in the paper. We include this new data in Fig. 4e-h”.

This experiment is not convincing. To examine the physiological consequence of the possible interaction of GABA with the M-channels on neuronal excitability (DRG or central neurons) is to record in the current-clamp configuration and in the presence of GABA_A and GABA_B blockers, the spike discharge with and without GABA.

In conclusion, the authors made a potentially interesting observation that M-channels could interact with GABA and may act as a sensor of the GABAergic milieu to regulate neuronal excitability.

Unfortunately, the experimental data provided in this revised manuscript yet do not support this claim.

-We have shown the effects of GABA in DRG neurons in the presence of canonical GABA_A and GABA_B receptor blockers, as previously requested. In the future, we look forward to studying the precise physiological roles in more detail and have begun the process of making transgenic mice to facilitate this. We feel this is for a follow-up paper as it will take well over a year to perform these studies properly, and the present manuscript contains many fundamental and highly novel findings that have been corroborated by extensive experimentation and controls, now in four different cell types.

Reviewer #3 (Remarks to the Author):

The authors satisfactorily revised the manuscript in response to my previous comments, and I have no more specific comments. I judge this ms is ready for publication.

-Thank you.

REVIEWERS' COMMENTS:

Reviewer #1 (Remarks to the Author):

Thanks for the rapid and detailed response to the previous critiques. I am satisfied with the paper.

I am concerned that the criticism related to tail current voltages is not correct - perhaps there is some confusion. The purpose of the tail current in an experiment like Figure 2c is to provide a simple index of the number of channels that open in the conditioning prepulse. The logic is that since the electrochemical driving force is constant in the tail currents, the only factor determining tail current magnitude is the number of channels that open in the prepulse. Within reason, it does not matter what tail current voltage is chosen (as long as a faithful measurement can be made shortly after the step to the tail voltage - this condition is easily satisfied in KV7 channels because they gate so slowly). Some groups may prefer to choose a voltage where channels are beginning to close, but this will have no meaningful effect on the conductance-voltage relationship extracted from this experiment.

I hope this is helpful in addition to the authors' response.

Reviewer #3 (Remarks to the Author):

I (Reviewer #3) checked the responses by the authors to the comments of Reviewers #1 and #2.

In my opinion, the authors' responses are full and sufficient. I agree with and support the authors' rebuttal to point #3 of Reviewer #2 in the following.

In the future, we look forward to studying the precise physiological roles in more detail and have begun the process of making transgenic mice to facilitate this. We feel this is for a follow-up paper as it will take well over a year to perform these studies properly, and the present manuscript contains many fundamental and highly novel findings that have been corroborated by extensive experimentation and controls, now in four different cell types.

I evaluate novel findings of this paper very high, and also judge this work is scientifically sound. Thus, I would like to recommend publication of this ms in "Nature Communications" in the present form, with a high priority.